



# A new airborne broadband radiometer system and an efficient method to correct thermal offsets

André Ehrlich[1],[★], Martin Zöger[2],[★], Andreas Giez[2], Vladyslav Nenakhov[2], Christian Mallaun[2], Rolf Maser[3], Timo Röschenthaler[3], Anna E. Luebke[1], Kevin Wolf[1],[a], Bjorn Stevens[4], and Manfred Wendisch[1]

[1]Leipzig Institute for Meteorology, Leipzig University , Germany
[2]German Aerospace Center, Flight Experiments, Oberpfaffenhofen, Germany
[3]enviscope GmbH, Frankfurt am Main, Germany
[4]*Max Planck* Institute for Meteorology, Hamburg, Germany
[a]now at: Institute *Pierre-Simon Laplace*, Sorbonne Université, Paris, France
[★]These authors contributed equally to this work.

**Correspondence:** André Ehrlich (a.ehrlich@uni-leipzig.de)

**Abstract.** The instrumentation of the High Altitude and Long Range (HALO) research aircraft is extended by the new Broadband AirCrAft RaDiometer Instrumentation (BACARDI) to quantify the radiative energy budget. Two sets of pyranometers and pyrgeometers are mounted to measure upward and downward solar (0.3–3 μm) and thermal-infrared (3–100 μm) irradiances. The radiometers are installed in a passively ventilated fairing to reduce the effects of the dynamic environment, e.g., fast

changes of altitude and temperature. The remaining thermal effects range up to $20\,\mathrm{W\,m^{-2}}$ for the pyranometers and $10\,\mathrm{W\,m^{-2}}$ for the pyrgeometers; they are corrected using an new efficient method that is introduced in this paper. Using data collected by BACARDI during a night flight, the thermal offsets are parameterized by the rate of change of the radiometer sensor temperatures. Applying the sensor temperatures instead of ambient air temperature for the parameterization provides a linear correction function ($200$–$600\,\mathrm{W\,m^{-2}\,K^{-1}\,s}$), that depends on the mounting position of the radiometer on HALO. Furthermore,

BACARDI measurements from the EUREC[4]A (Elucidating the role of clouds-circulation coupling in climate) field campaign are analyzed to characterize the performance of the radiometers and to evaluate all corrections applied in the data processing. Vertical profiles of irradiance measurements up to 10 km altitude show that the thermal offset correction limits the bias due to temperature changes to values below $10\,\mathrm{W\,m^{-2}}$. Measurements with BACARDI during horizontal, circular flight patterns in cloud-free conditions demonstrate that the common geometric attitude correction of the solar downward irradiance provides

reliable measurements in this typical flight sections of EUREC[4]A, even without active stabilization of the radiometer.

## 1 Introduction

Measurements of solar and thermal-infrared irradiance are important to quantify the radiative impact of atmospheric components and surface properties on the Earth's radiative energy budget and to quantify their relevance for climate change. Ground-based observations of the broadband upward and downward irradiances are routinely performed within the Baseline Surface

Radiation Network (BSRN) at locations distributed over the entire globe (Driemel et al., 2018). These observations were used



in a variety of studies, e.g., characterizing the climatology of cloud radiative effects for example by Shupe and Intrieri (2004). However, BSRN observations are limited to fixed land locations representing a local environment, e.g., surface albedo or temperature regime. Observations over ocean are obtained only from a few research ships and buoys (Kalisch and Macke, 2012; Colbo and Weller, 2009). Instead, airborne or space-borne observations resolve the spatial distribution of the radiative energy budget, which is strongly affected by the heterogeneity of the surface albedo, surface temperature and clouds (Stapf et al., 2020, 2021). While satellite estimates of the irradiances at top of atmosphere require radiative transfer simulations, airborne observations provide direct measurements of the upward and downward, solar and thermal-infrared irradiance. Furthermore, radiative processes such as cloud top cooling or aerosol layer warming need to be quantified to understand the influence of radiative processes on atmospheric dynamics (e.g., Wendisch et al., 2008; Simpfendoerfer et al., 2019). These quantities are derived from profiles of net (downward minus upward) irradiances, which can by measured directly only by airborne observations (Bucholtz et al., 2010) or from balloon and helicopter platforms (Egerer et al., 2019; Siebert et al., 2021).

Broadband irradiances $F$ are measured by radiometer, in particular pyranometer (solar, 0.3–3 µm) and pyrgeometer (thermal-infrared, 3–100 µm) respectively. The measurement principle of most common radiometers, as discussed here, is based on thermopile sensors. Some radiometers use photo-diode sensors, which are sensitive only to a limited spectral range, while thermopile sensors in general detect the entire spectral range of electromagnetic radiation. To define the wavelength selectivity of a thermopile radiometer and to protect the sensor from environmental impacts, the sensor is capped by a dome. Special materials, e.g., quartz glass, silicon, as well as filter coatings guarantee a relatively constant sensitivity of the instrument over the desired spectral range (Gröbner and Los, 2007). The overall performance of broadband radiometers is determined by the radiometric calibration accuracy, dome spectral transmissivity, angular response, direct solar heating, dome temperature effects, and long-term measurement stability (e.g., Philipona et al., 1995, 2001; Wendisch and Brenguier, 2013; Gröbner et al., 2014).

The combination of a thermopile sensor and an optical filter dome can affect the thermal equilibrium of the entire instrument, and thus bias the measurements, especially when operating the radiometer on aircraft, where fast radiation and temperature changes may occur (e.g., Albrecht et al., 1974; Saunders et al., 1992). Thermopile radiometers typically have an inertia of a few seconds to changes in radiation. However, Curry and Herman (1985) and Foot (1986) showed that thermal equilibrium, especially during rapid ascents and descents, may be reached only after several minutes. This effect is caused by a differential change of the temperatures between the dome and the thermopile sensor, which was estimated by Philipona et al. (1995) with up to $\pm 1.5\,°\mathrm{C}$. To minimize these uncertainties, Philipona et al. (1995) suggested adding two additional temperature sensors in the dome and parameterizing the irradiance bias. However, commercially available broadband radiometers, which are built for ground-based operation, do not include these temperature sensors and require careful post-processing (Ehrlich and Wendisch, 2015).

Airborne measurements, especially of the downward solar irradiance, are also affected by the aircraft attitude when the radiometers are fixed to the aircraft fuselage (Wendisch et al., 2001). By definition, the atmospheric irradiance refers to a horizontally aligned surface, which is not maintained by the radiometer sensor during pitch and roll aircraft movements. Depending on the solar zenith angle, Wendisch et al. (2001) calculated that a misalignment of $\pm 1°$ already results in an offset of up to 3 % in the downward solar irradiance for a solar zenith angle of 60°. Actively stabilized pyranometer, such as those





proposed by Wendisch et al. (2001) and Bucholtz et al. (2008), can minimize such uncertainties, but these techniques are complex and expensive and not applicable to all aircraft installations. A post-correction as suggested by, e.g., Bannehr and Schwiesow (1993) and Boers et al. (1998), is limited to the direct solar component of the incoming radiation (cloud-free conditions) and depends on the accuracy of the estimation of the fraction of direct solar radiation, the characterization of the

pyranometer mounting, and the measurement of the aircraft attitude.

This attitude correction requires synchronized pyranometer and aircraft attitude measurements, which may not be given due to the slow response of the broadband radiometer (Freese and Kottmeier, 1998). As shown by Ehrlich and Wendisch (2015), characterizing the radiometer time response and reconstructing the measurement time series significantly improves the performance of airborne radiometers and helps to analyze the radiation field in complex cloud and surface conditions (Egerer

et al., 2019; Stapf et al., 2020).

Given these known issues, airborne measurements of broadband solar and thermal-infrared irradiance are delicate and require a proper setup of the radiometers on the aircraft as well as careful post-processing aimed at correcting most of the inevitable effects. Here, a new radiometer package, the Broadband AirCrAft RaDiometer Instrumentation (BACARDI) installed on the High Altitude And Long Range (HALO) research aircraft operated by the German Aerospace Center (Deutsches Luft und

Raumfahrtzentrum, DLR), is introduced. BACARDI comprises of a set of two Kipp and Zonen pyranometers (CMP22) and pyrgeometers (CGR4) that are mounted in a fixed position to the aircraft fuselage. The housing and mounting is constructed to minimize thermal effects. However, thermal offsets remain and therefore a novel approach to correct for them is developed. To illustrate the basis of the correction, Section 2 gives a review of the theory of the broadband radiometer radiative budget. The radiometer characteristics, data acquisition specification, and the instrument design, including the aircraft specific instrument

mounting and shielding, is described in Section 3. All basic corrections including the radiometric calibration, the reconstruction of the time response, and the attitude correction of the solar downward irradiance are specified in Section 4. Based on a dedicated calibration flight, a novel approach to correct the thermal offset of the radiometer in rapidly changing temperature conditions is developed. Section 5 outlines the correction approach and the application to measured vertical profiles of solar and thermal-infrared irradiance. The overall performance of BACARDI is tested using measurements during the EUREC$^4$A

(Elucidating the role of clouds-circulation coupling in climate) field campaign (Bony et al., 2017; Stevens et al., 2021). Measurements of heating rate profiles and the consistency of measurements during circular flight pattern is analyzed by comparison to radiative transfer simulations. The key benefits of the new system are summarized in Section 7.

## 2    The radiative energy budget of broadband radiometer

### 2.1    Basics

Broadband radiometers, which are based on thermopile sensors, use the temperature increase of the illuminated receiver area compared to a shaded reference area as the primary measure. Based on the thermopile sensitivity $\alpha$ in units of $\mathrm{K\,V^{-1}}$ (Seebeck effect), the resulting temperature difference $\Delta T$ between the sensor surface temperature $T_\mathrm{s}$ and the reference area temperature





$T_{\text{ref}}$ generates a detectable voltage $U_{\text{th}}$, which is used to compute the irradiance,

$$U_{\text{th}} = \frac{1}{\alpha} \cdot \Delta T = \frac{1}{\alpha} \cdot (T_{\text{s}} - T_{\text{ref}}). \tag{1}$$

In most cases, the temperature of the reference area is measured with a standard temperature sensor, e.g., Pt-100 or thermistor. The sensor itself absorbs the incoming irradiance $F_{\text{in}}$, and also emits radiation $F_{\text{out}}$ in the thermal-infrared wavelength range. In case of pyrgeometer, the emission is a major issue. From a simplified energy budget of the thermopile sensor, the total effect of the net irradiance $F_{\text{net,dyn}}$ on the sensor can be described by,

$$F_{\text{net,dyn}} = F_{\text{in}} - F_{\text{out}} = C \cdot \frac{\partial T_{\text{s}}}{\partial t} + K \cdot \Delta T, \tag{2}$$

with $C$ the heat capacity of the sensor surface in units of $\mathrm{J\,m^{-2}\,K^{-1}}$ and $K$ the thermopile thermal conductance in units of $\mathrm{W\,m^{-2}\,K^{-1}}$.

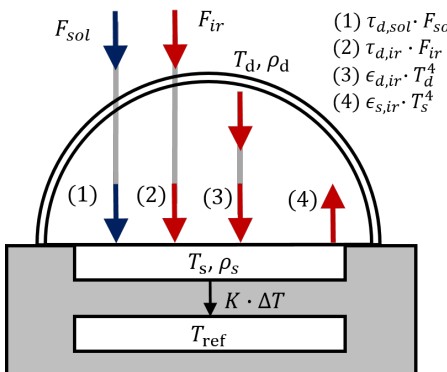

**Figure 1.** Simplified radiative energy budget of a broadband radiometer composed of: (1) the transmitted solar irradiance, (2) the transmitted thermal-infrared irradiance, (3) the irradiance emitted by the dome, and (4) the irradiance emitted by the sensor. Blue arrows indicate solar radiation, red arrows are used for thermal-infrared radiation.

## 2.2   Thermal equilibrium

Assuming thermal equilibrium, $\partial T_{\text{s}}/\partial t = 0$, Eq. (2) reduces to the net irradiance $F_{\text{net,stat}}$ in static conditions:

$$F_{\text{net,stat}} = F_{\text{in}} - F_{\text{out}} = K \cdot \Delta T, \tag{3}$$

Following the simplified radiative budget of a broadband radiometer as shown in Fig. 1, the standard formulation of the calibration equation of broadband radiometers in radiative equilibrium is postulated (e.g., Fairall et al., 1998; Ji and Tsay, 2000). The incoming and outgoing irradiance $F_{\text{in}}$ and $F_{\text{out}}$ at the sensor surface can be expressed independently by:

$$F_{\text{in}} = \tau_{\text{d,sol}} \cdot F_{\text{sol}} + \tau_{\text{d,ir}} \cdot F_{\text{ir}} + \epsilon_{\text{d,ir}} \cdot \sigma \cdot T_{\text{d}}^4 + \rho_{\text{d}} \cdot F_{\text{out}} \tag{4}$$

$$F_{\text{out}} = \epsilon_{\text{s,ir}} \cdot \sigma \cdot T_{\text{s}}^4 + \rho_{\text{s}} \cdot F_{\text{in}}, \tag{5}$$





with $\tau_{\mathrm{d,sol}}$ the solar transmissivity of the dome, $T_{\mathrm{d}}$ the temperature of the dome, and $\tau_{\mathrm{d,ir}}$ and $\epsilon_{\mathrm{d,ir}}$ the thermal-infrared transmissivity and emissivity of the dome. The sensor is characterized by the sensor emissivity $\epsilon_{\mathrm{s,ir}}$ and temperature $T_{\mathrm{s}}$. $\sigma$ is the Stefan-Boltzmann constant. The sensor reflects the incoming irradiance $F_{\mathrm{in}}$ with the reflectivity $\rho_{\mathrm{s}}$, while the dome can reflect the outgoing irradiance $F_{\mathrm{out}}$ with a reflectivity $\rho_{\mathrm{d}}$. Assuming that $\rho_{\mathrm{s}} \cdot \rho_{\mathrm{p}} \ll 1$ and replacing $T_{\mathrm{s}}$ by $T_{\mathrm{ref}}$ using the following assumption for the polynomial of 4th order,

$$T_{\mathrm{s}}^4 = (T_{\mathrm{ref}} + \alpha \cdot U_{\mathrm{th}})^4 \approx T_{\mathrm{ref}}^4 + 4 \cdot T_{\mathrm{ref}}^3 \cdot \alpha \cdot U_{\mathrm{th}}, \tag{6}$$

the equations finally can be resolved for the thermal-infrared and solar incident irradiance:

$$F_{\mathrm{ir}} = U_{\mathrm{th}} \cdot \frac{\alpha \cdot K}{\epsilon_{\mathrm{s,ir}} \cdot \tau_{\mathrm{d,ir}}} \cdot \left( 1 + \frac{4 \cdot \sigma \cdot T_{\mathrm{ref}}^3}{K \cdot (\epsilon_{\mathrm{d,ir}}/\tau_{\mathrm{d,ir}} + 1) \cdot (\epsilon_{\mathrm{s,ir}} \cdot \tau_{\mathrm{d,ir}})} \right) + \sigma \cdot T_{\mathrm{ref}}^4 + \frac{\epsilon_{\mathrm{d,ir}}}{\tau_{\mathrm{d,ir}}} \cdot \sigma \cdot \left( T_{\mathrm{ref}}^4 - T_{\mathrm{d}}^4 \right) - F_{\mathrm{sol}} \cdot \frac{\tau_{\mathrm{d,sol}}}{\tau_{\mathrm{d,ir}}}, \tag{7}$$

$$F_{\mathrm{sol}} = U_{\mathrm{th}} \cdot \frac{\alpha \cdot K}{\epsilon_{\mathrm{s,ir}} \cdot \tau_{\mathrm{d,sol}}} \cdot \left( 1 + \frac{4 \cdot \sigma \cdot T_{\mathrm{ref}}^3}{K \cdot (\epsilon_{\mathrm{d,ir}} + \tau_{\mathrm{d,ir}}) \cdot (\epsilon_{\mathrm{s,ir}} \cdot \tau_{\mathrm{d,sol}})} \right) + \frac{\epsilon_{\mathrm{d,ir}}}{\tau_{\mathrm{d,sol}}} \cdot \sigma \cdot \left( T_{\mathrm{ref}}^4 - T_{\mathrm{d}}^4 \right) + \frac{\tau_{\mathrm{d,ir}}}{\tau_{\mathrm{d,sol}}} \cdot \left( \sigma \cdot T_{\mathrm{ref}}^4 - F_{\mathrm{ir}} \right). \tag{8}$$

The last term in Eq. (7) and Eq. (8) the so called longwave and shortwave leakage, is only a function of the incident irradiance and the ratio between solar and thermal-infrared transmissivity of the dome, which are determined by material properties of the dome and/or filter coating. There is evidence for the existence of such errors due to spectral imperfections of the dome, and possible corrections were developed (e.g., Pascal and Josey, 2000). By careful selection of the dome material and coating (low $\tau_{\mathrm{d,sol}}$ and high $\tau_{\mathrm{d,ir}}$ for pyrgeometers and high $\tau_{\mathrm{d,sol}}$ and low $\tau_{\mathrm{d,ir}}$ for pyranometers), this error can be minimized or even neglected as confirmed by long-term comparison of different shaded and illuminated pyrgeometers (Meloni et al., 2012; Kipp and Zonen, 2014). For pyranometers, the longwave leakage is correlated to the net thermal-infrared irradiance measured by a pyrgeometer and thus can hardly be distinguished from the thermal dome offset.

Neglecting this leakage effect, Eq. (7) and Eq. (8) can be reduced to the commonly known formulas (Philipona et al., 1995). For the thermal-infrared irradiance measured by pyrgeometer it is:

$$
\begin{aligned}
F_{\mathrm{ir}} &= A_1 \cdot U_{\mathrm{th}} \cdot \left( 1 + A_2 \cdot \sigma \cdot T_{\mathrm{ref}}^3 \right) + \sigma \cdot T_{\mathrm{ref}}^4 - A_3 \cdot \sigma \cdot \left( T_{\mathrm{d}}^4 - T_{\mathrm{ref}}^4 \right), \\
A_1 &= \frac{\alpha \cdot K}{\epsilon_{\mathrm{s,ir}} \cdot \tau_{\mathrm{d,ir}}}, \\
A_2 &= \frac{4}{K \cdot (\epsilon_{\mathrm{d,ir}}/\tau_{\mathrm{d,ir}} + 1) \cdot (\epsilon_{\mathrm{s,ir}} \cdot \tau_{\mathrm{d,ir}})}, \\
A_3 &= \frac{\epsilon_{\mathrm{d,ir}}}{\tau_{\mathrm{d,ir}}}
\end{aligned}
\tag{9}
$$

with the parameter $A_1$, $A_2$, $A_3$ summarizing the instrument characteristics. If the temperature dependence of the thermopile sensitivity, $A_2 \cdot \sigma \cdot T_{\mathrm{ref}}^3$, is compensated electronically within the radiometer, the first term of Eq. (9) can further be reduced leading to the formulation by Albrecht et al. (1974):

$$
\begin{aligned}
F_{\mathrm{ir}} &= a_{\mathrm{ir}} \cdot U_{\mathrm{th}} + \sigma \cdot T_{\mathrm{ref}}^4 - b_{\mathrm{ir}} \cdot \sigma \cdot \left( T_{\mathrm{d}}^4 - T_{\mathrm{ref}}^4 \right), \\
a_{\mathrm{ir}} &= A_1 \cdot \left( 1 + A_2 \cdot \sigma \cdot T_{\mathrm{ref}}^3 \right), \\
b_{\mathrm{ir}} &= A_3,
\end{aligned}
\tag{10}
$$





with $a_{\mathrm{ir}}$ the adjusted pyrgeometer thermopile sensitivity in units $\mathrm{W\,m^{-2}\,V^{-1}}$ and $b_{\mathrm{ir}}$ the adjusted pyrgeometer dome factor.

The last term, also known as the window heating offset, corrects for a thermal imbalance between dome and sensor surface mainly caused by solar radiative heating of the dome. The dome factor $b_{\mathrm{ir}}$ in Eq. (10) theoretically defines the ratio of thermal-infrared emissivity $\epsilon_{\mathrm{d,ir}}$ to transmissivity of the dome $\tau_{\mathrm{d,ir}}$. Ji and Tsay (2000) showed that the dome factor, experimentally determined from a black body calibration of the instrument, yields significantly higher values than expected from theory. Only by using data obtained in thermal equilibrium, the theory is fulfilled. This indicates that the commonly used higher dome factor

implies non-equilibrium effects. Optimizing the thermal design of the radiometer can reduce the window heating offset such that a dome temperature measurement can be omitted and no dome factor is needed (Meloni et al., 2012; Gröbner et al., 2014; Kipp and Zonen, 2014).

Applying a similar transformation, the solar irradiance measured by pyranometers Eq. (8) reduces to:

$$
\begin{aligned}
F_{\mathrm{sol}} =& a_{\mathrm{sol}} \cdot U_{\mathrm{th}} - b_{\mathrm{sol}} \cdot \sigma \cdot \left(T_{\mathrm{d}}^4 - T_{\mathrm{ref}}^4\right), \\
a_{\mathrm{sol}} =& \frac{\alpha \cdot K}{\epsilon_{\mathrm{s,ir}} \cdot \tau_{\mathrm{d,sol}}} \cdot \left(1 + \frac{4 \cdot \sigma \cdot T_{\mathrm{ref}}^3}{K \cdot (\epsilon_{\mathrm{d,ir}} + \tau_{\mathrm{d,ir}}) \cdot (\epsilon_{\mathrm{s,ir}} \cdot \tau_{\mathrm{d,sol}})}\right), \\
b_{\mathrm{sol}} =& \frac{\epsilon_{\mathrm{d,ir}}}{\tau_{\mathrm{d,sol}}}.
\end{aligned}
\tag{11}
$$

The adjusted pyranometer thermopile sensitivity $a_{\mathrm{sol}}$ in units $\mathrm{W\,m^{-2}\,V^{-1}}$ includes the weak temperature dependence of the thermopile as defined in theory by Eq. (8), which can often be compensated by the construction of the radiometer or determined in extended laboratory calibrations.

The pyranometer thermal dome effect is scaled with the dome factor $b_{\mathrm{sol}}$ and the temperature difference between dome and sensor. This effect is often called the zero or dark offset since it is mainly caused by radiative cooling of the dome and is best

visualized as a negative offset during night measurements in the absence of solar irradiance. A second dome with high thermal conductivity, e.g., quartz, in good thermal contact with the instrument housing can reduce this error to a few $\mathrm{W\,m^{-2}}$ (Philipona, 2002; Reda et al., 2005; Kipp and Zonen, 2016). Ventilation of the dome can further reduce the zero offset (Michalsky et al., 2017). If the thermal dome effect cannot be neglected, available corrections methods are applied. These have been developed based on either an additional dome temperature measurement or the simultaneous measurement of the net thermal-infrared

irradiance by a pyrgeometer (e.g., Bush et al., 2000; Haeffelin et al., 2001; Dutton et al., 2001).

### 2.3 Dynamic environment – no thermal equilibrium

The assumption of thermal equilibrium is valid for standard ground-based measurements with slowly varying environmental conditions. However, if the radiometers are subject to fast temperature changes like during airborne measurements, e.g., during ascents and descents, the slow adjustment of the sensor temperature (first term in Eq. 2) needs to be considered. An offset

voltage $\Delta U_{\mathrm{s}}$ and offset irradiance $\Delta F_{\mathrm{s}}$, respectively, will be generated by the thermal lag between the reference and the sensor, which is initiated by the thermal conductance and capacity of the sensor.





Replacing the sensor temperature $T_\mathrm{s}$ in Eq. (2) by the reference temperature, $T_\mathrm{s} = T_\mathrm{ref} + \Delta T$, the thermal reaction of the the sensor to an outside temperature change can be described by:

$$F_\mathrm{net,dyn} = C \cdot \left( \frac{\partial T_\mathrm{ref}}{\partial t} + \frac{\partial \Delta T}{\partial t} \right) + K \cdot \Delta T. \tag{12}$$

The assumption of $\Delta T \ll T_\mathrm{ref}$ leads to the sensor thermal offset $\Delta F_\mathrm{s}$ defined as the difference between the net irradiance in static $F_\mathrm{net,stat}$ and dynamic conditions $F_\mathrm{net,dyn}$:

$$\Delta F_\mathrm{s} = F_\mathrm{net,stat} - F_\mathrm{net,dyn} = -C \cdot \frac{\partial T_\mathrm{ref}}{\partial t}. \tag{13}$$

This error correction term for dynamic temperature changes is proportional to the time derivative of the reference temperature. $\Delta F_\mathrm{s}$ often is called 'zero offset B' or 'zero offset due to temperature change' and is mostly specified by the instrument

manufacturer for a fixed temperature change of $5\,\mathrm{K\,h^{-1}}$. However, during airborne observation, especially during ascents and descents, faster temperature changes in the order of $\mathrm{K\,min^{-1}}$ occur.

A similar behavior is expected for the dome, leading to a slow adjustment of the dome temperature $T_\mathrm{d}$. Due to the different thermal properties of dome and sensor, the thermal offset in both parts do not compensate. The dynamic dome effect $\Delta F_\mathrm{d}$ can then be expressed as:

$$\Delta F_\mathrm{d} = \sigma \cdot (T_\mathrm{ref}^4 - (T_\mathrm{ref} + \Delta T_\mathrm{d})^4). \tag{14}$$

As indicated by Bush et al. (2000), the temperature difference of the dome $\Delta T_\mathrm{d}$, depends linearly on the temporal change of $T_\mathrm{ref}$:

$$\Delta T_\mathrm{d} = \gamma \cdot \frac{\partial T_\mathrm{ref}}{\partial t}, \tag{15}$$

with the coefficient $\gamma$ in units of s characterizing the relationship. Assuming $\Delta T_\mathrm{d} \ll T_\mathrm{ref}$ and approximating the 4th order

polynomial similar to Eq. (6), the dome effect reduces to:

$$\Delta F_\mathrm{d} = 4 \cdot \sigma \cdot \gamma \cdot T_\mathrm{ref}^3 \cdot \frac{\partial T_\mathrm{ref}}{\partial t}. \tag{16}$$

Adding both effects (Eq. 13 and Eq 16), the total thermal offset $\Delta F$ is described by:

$$\Delta F = \Delta F_\mathrm{s} + \Delta F_\mathrm{d} = \left( 4 \cdot \sigma \cdot \gamma \cdot T_\mathrm{ref}^3 - C \right) \cdot \frac{\partial T_\mathrm{ref}}{\partial t} = \beta \cdot \frac{\partial T_\mathrm{ref}}{\partial t}. \tag{17}$$

This theory indicates that the offset can be linearly parameterized by the change rate of the sensor reference temperature provid-

ing the thermal offset corrections coefficient $\beta$ in units of $\mathrm{W\,m^{-2}\,K^{-1}\,s}$. During data post-processing, such a parameterization can be applied to correct the irradiance measurements in high dynamic conditions.

## 3  Design of BACARDI for operation on HALO

### 3.1  Broadband radiometer

For the measurements of upward and downward broadband irradiance, $F^\downarrow$ and $F^\uparrow$, separated into the solar and thermal-

infrared spectral range, BACARDI combines two sets of Kipp and Zonen pyranometers (CMP22) and pyrgeometers (CGR4).





The CMP22 pyranometers detect radiation in the wavelengths range of $0.2 - 3.6\,\mu m$, which covers almost the entire solar spectral range (Kipp and Zonen, 2016). The CGR4 pyrgeometer is sensitive to wavelengths between $4.5 - 42\,\mu m$ covering a large fraction of thermal-infrared radiation (Kipp and Zonen, 2014). Both radiometers use thermopile sensors providing a sensitivity in the range of $10\,\mu V\,(W\,m^{-2})^{-1}$. The radiometric calibration of the radiometers, which refers to the entire solar and thermal-infrared spectral range, respectively, is repeated regularly by the manufacturer a few months in advance of a HALO measurement campaign. The radiometers are calibrated as secondary standard (Class A) through comparison with a reference instrument traceable to the World Radiation Center. For the pyranometers, this comparison is done in the laboratory, and for the pyrgeometers, the comparison is performed outside under mainly cloud-free conditions during night time. Additionally, for both radiometers the temperature dependence of the thermopile sensitivity is determined within a climate chamber for the temperature range of -40 to $50\,°C$. Calibration uncertainties typically range below $1\,\%$ for the CMP22 pyranometers and $5\,\%$ for the CGR4 pyrgeometers. The temperature dependence of the thermopiles does not exceed $0.5\,\%$ for a wide temperature range (-30 to $50\,°C$). To track the sensor temperature, each radiometer is equipped by the manufacturer with a platinum (Pt-100) resistance thermometer.

The respective quartz and silicon domes function as wavelength band pass filters and are characterized by a cosine response, which is less than $1\,\%$ off from theory over the entire $180°$ field-of-view (Kipp and Zonen, 2014, 2016). The optimized thermal design of both radiometers reduces the window heating offset to less than $4\,W\,m^{-2}$ and makes them suited for aircraft operation. However, the time response of the radiometer needs to be considered for airborne measurements. As specified by the manufacturer, the CMP22 pyranometer typically reacts quicker with a $1/e$ ($63\,\%$ adjustment) response time of about $2\,s$, while the CGR4 pyrgeometer is characterized by a response time on the order of $6\,s$.

## 3.2 Electronics and data acquisition

The CMP22 and CGR4 radiometers do not contain any internal signal conditioning and only provide a low voltage (a few $\mu V\,(W\,m^{-2})^{-1}$) thermopile signal and a 4-wire Pt-100 temperature signal.

To avoid mitigating of electromagnetic noise, the wiring of the low voltage signal is as short as possible and a signal conditioning unit is used. This unit is placed on the radiometer mounting plate inside the fuselage, where it is protected by an electromagnetic compatibility shielding metal box. The signal conditioning is based on isolated Dataforth 8B modules that are plugged into a backplane. The 8B30-02 module with an input range of $\pm 50\,mV$ is used for amplification of the thermopile signals up to $\pm 5\,V$, with an accuracy of $0.05\,\%$. The 4-wire Pt-100 resistance is translated into a voltage ($0 - 5\,V$) by the 8B35-01 module, which covers the temperature range $\pm 100\,°C$ with an uncertainty of $\pm 0.2\,°C$.

The output voltage signals have a bandwidth of $3\,Hz$ and are recorded by the HALO basic data acquisition system BAHAMAS (BAsic HAlo Measurement And Sensor system, Krautstrunk and Giez, 2012) with a $10\,Hz$ data rate and 18-bit resolution. A signal path calibration is performed after aircraft installation of the radiometers, which includes all wiring, connectors, electronics, and the data acquisition. For the calibration, the radiometers are replaced by either a high precision constant voltage source (Burster 4463) to simulate the thermopile output, or a high precision resistance decade (Burster 1427) to simulate the Pt-100. Both calibration references are set to values covering the operating range of the radiometer and Pt-100 thermometer



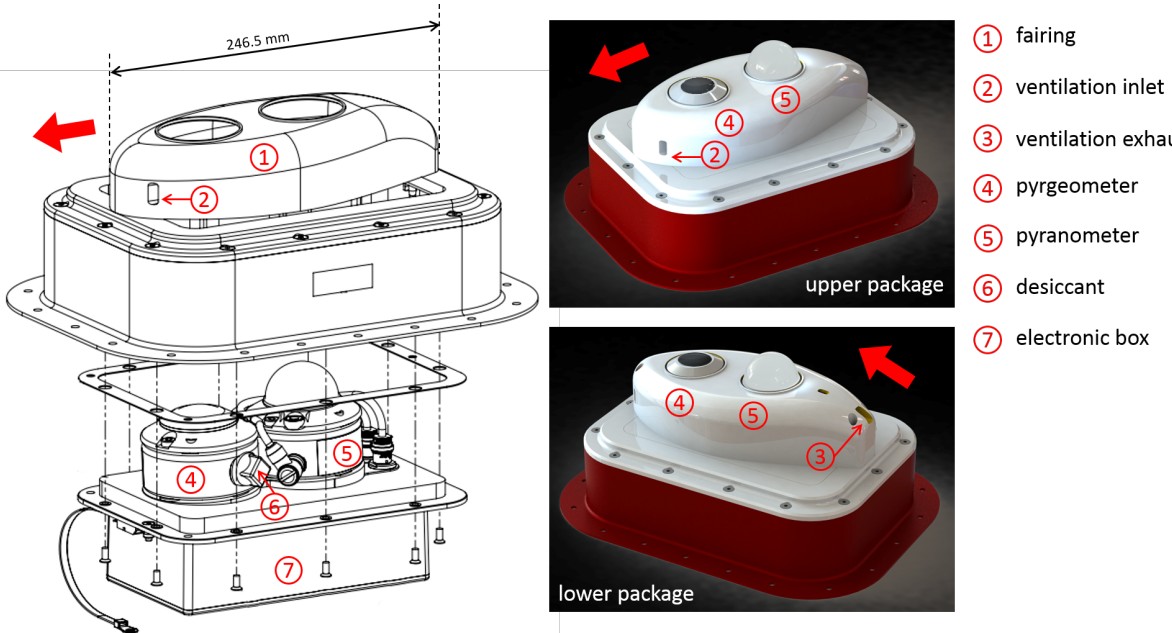

**Figure 2.** Drawing and visualization of the BACARDI sensor packages illustrating the main components: fairing (1) with ventilation inlet (2) and exhaust (3), pyrgeometer (4), pyranometer (5), desiccant cartridge (6), and electronic box (7). The red arrow indicates the flight direction.

by a computer controlled calibration routine. The calibration factors are implemented in the first post-processing step of the BACARDI raw data.

### 3.3 Mounting on HALO and fairing

The integration of BACARDI on HALO uses the standard 10" x 7" fuselage apertures. Because HALO is equipped with 4 upper and 6 lower central apertures, some flexibility in installation depending on the layout of the actual scientific instrumentation
is given. A drawing and visualization of one BACARDI sensor package is shown in Fig. 2. The mounting plates, to which the radiometers are attached, compensate for the mean pitch angle of the HALO aircraft, which amounts to about -3° in normal flight conditions. To reduce the cable length between the radiometers and the electronics (amplifier and Pt-100 conditioner), the electronics housing is attached to the mounting plate on the opposite side of the radiometers inside the fuselage.

The radiometers of BACARDI are in an aerodynamic fairing to minimize the environmental influence on the radiation
measurement by, e.g., ice aggregation or water droplet impact and heating by solar radiation. To minimize aerodynamically induced temperature gradients across the instrument, a passive ventilation of the fairing is implemented to keep the instruments close to thermal equilibrium with its surrounding environment. The ventilation is designed to divert the main airflow containing droplets or particles around the radiometer housings. The fairing exhaust acts as a water drain and avoids entrapment of water inside the fairing. Thus, the design of the fairing for the upward looking radiometers slightly differs from that for the downward
looking radiometers.





Figure 3 shows measurements of all four sensor temperatures compared to the ambient temperature measured on HALO. In general, the sensor temperatures are higher, especially in cold conditions, due to low heat transfer in the rather low-density air and the heat conduction from the cabin. Temperature adjustments to changes in ambient temperature (change in altitude) significantly lag in time and may lead to thermal offsets as discussed in Section 2.3. This is most prominently indicated

by the hysteresis between ascent (upper branch) and descent (lower branch) in Fig. 3. However, comparing only the sensor temperatures, the differences are larger between the pyranometers and pyrgeometers than between the upper and lower setup. This indicates that temperature adjustments are rather a matter of the internal sensor housing than caused by the ventilation within the fairing.

To enables maintenance work, e.g., the change of desiccant cartridges, and the signal calibration, easy access to the ra-

diometers is considered necessary. Therefore, the upward looking radiometers can be detached from inside the cabin without removing the fairing, whereas for the downward looking radiometers, it is sufficient to dismount the fairing.

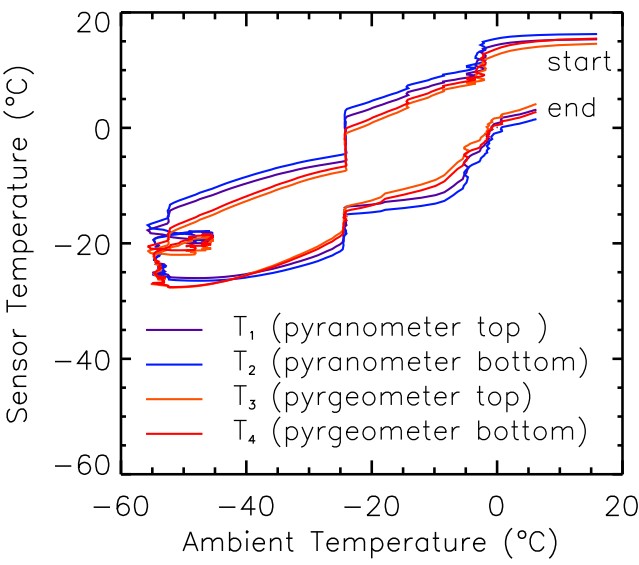

**Figure 3.** Radiometer sensor temperatures compared to the ambient air temperature measured during the EUREC[4]A flight on 22 January 2020 (Flight ID HALO-0122).

## 4 Basic corrections

### 4.1 Temperature dependence of thermopile sensitivity

The calibration of the pyranometers and pyrgeometers provided by the manufacturer include tracking changes in the thermopile

sensitivity with changing instrument temperatures. For the CMP22 pyranometers, the change of the sensitivity is estimated in





the range of $\pm 0.3\,\%$ for the temperature range between -20 and $50\,^{\circ}$C. Significant lower sensitivities of up to $-2\,\%$ are registered when temperatures reach $-40\,^{\circ}$C. For the CGR4 pyrgeometers, lower differences in the sensitivity are reported. Here, deviations do not exceed $\pm 0.5\,\%$, with the largest positive biases observed for the lowest and highest temperatures and slight negative offsets in between.

Figure 4a shows the sensor temperatures of all four radiometers and the ambient static air temperature for the EUREC[4]A flight on 22 January 2020 (Flight ID HALO-0122). Except for takeoff and landing, the flight altitude and, thus, the ambient temperature changed only for one flight section when HALO climbed to 13.5 km altitude. At cruising altitude, minimum ambient temperatures down to $-60\,^{\circ}$C were observed. However, due to thermal conduction from the aircraft, the sensor temperatures remained significantly higher and did not reach $-40\,^{\circ}$C, the lower boundary of the calibration certificate. The same holds for
all other flights during EUREC[4]A. In other environments, e.g., Arctic conditions, where low temperatures are reached at lower altitudes with higher air densities, extending the calibration to lower temperatures needs to be considered.

   The effect of the temperature dependence on the sensor sensitivities is shown in Fig. 4b and c. The changes in the sensor temperature are well documented, and the radiometric calibration is adjusted by up to $1\,\%$ ($0.1\,\mu$V W$^{-1}$ m$^2$). Converted into irradiance, this corresponds to a maximum correction of $5\,$W m$^{-2}$ for the downward solar irradiance during local solar noon
(16 UTC) when the Sun is high. Due to the lower thermal-infrared irradiances, the differences here are one magnitude lower.

## 4.2   Correction of sensor response time

The response times $\tau_\mathrm{r}$ of the CMP22 and CGR4 radiometers provided by the manufacturer are evaluated by measurements during a test flight in cloud-free conditions. The response time of the upward looking pyranometers is determined by the cross-correlation between the measured downward irradiance and the aircraft attitude angles, assuming that the aircraft attitude is
recorded instantaneously by the GPS-aided inertial navigation system. A $1/e$ ($63\,\%$ adjustment) response time of $\tau_\mathrm{r} = 1.2\,$s, which is slightly lower than reported by the manufacturer, is obtained. The same response time is assumed for the downward looking pyranometer.

   The response times of the CGR4 pyrgeometers are extensively characterized by Ehrlich and Wendisch (2015) in a laboratory study with a reported $\tau_\mathrm{r}$ of about 3 s. EUREC[4]A measurements of flight sections with sharp turns are used to validate the $\tau_\mathrm{r}$ of
the BACARDI pyrgeometers. During the turns, the upward looking radiometer partly observed the warmer lower hemisphere, which caused a sudden increase of the upward irradiance. Based on a detailed analysis of this systematic change, the response time is adjusted to $3.3\,$s.

   The inertia of the measured irradiances caused by these response times are corrected following the deconvolution method proposed by Ehrlich and Wendisch (2015). To minimize numerical effects of the deconvolution at sharp gradients and the sensor
noise, a cut-off frequency of $0.6\,$Hz and a moving average filter with $0.5\,$s window length are applied to the reconstruction of the pyranometer measurements. For the pyrgeometers, a slightly lower cut-off frequency of $0.5\,$Hz and a longer window length of 2 s are chosen.

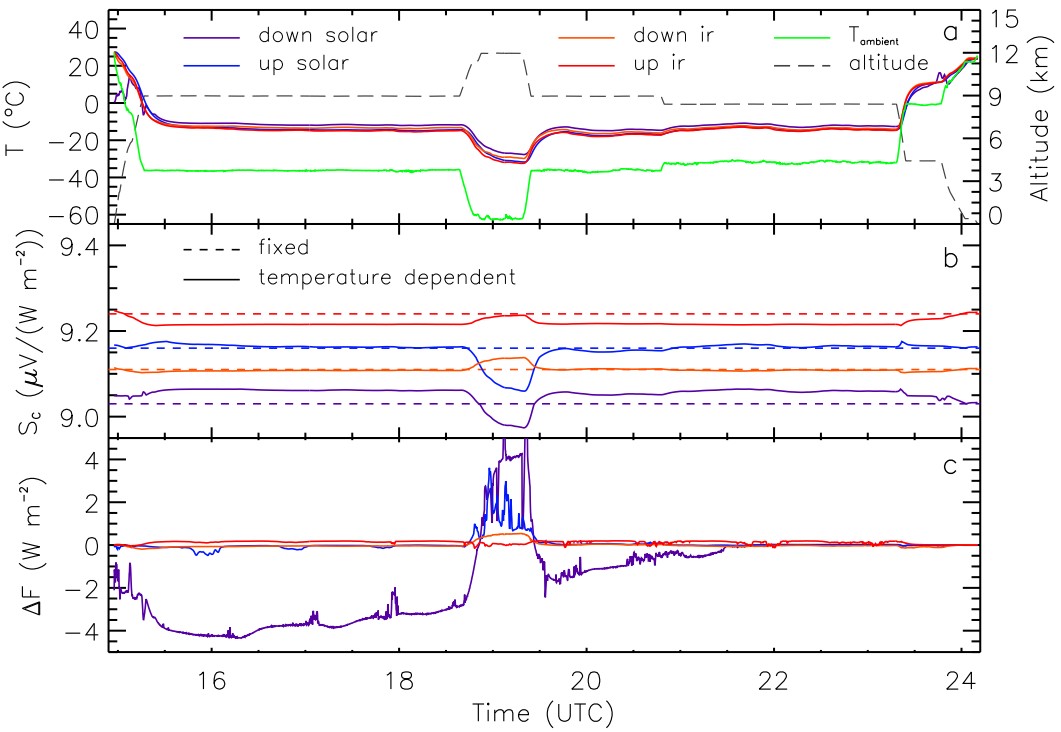

**Figure 4.** Time series of the ambient air and all radiometer sensor temperatures (**a**), the basic and corrected thermopile sensitivities of all radiometers (**b**), and the corrected bias of the irradiance due to the temperature dependence of the thermopile sensitivities (**c**) for the EUREC[4]A flight on 22 January 2020 (Flight ID HALO-0122). The flight altitude of HALO is given in panel (**a**).

## 4.3 Attitude correction of downward solar irradiance

BACARDI is fixed to the aircraft fuselage and does not actively align to the horizontal plane. Therefore, the measurements
are affected by the aircraft attitude. Except for turns, changes in the roll and pitch angles of HALO typically do not exceed $\pm 1°$, limiting the alignment error (Wendisch et al., 2001). For the downward solar irradiance, a post-correction following the approach by Bannehr and Schwiesow (1993) and Boers et al. (1998) is applied. This correction is possible only for the direct solar component. Therefore, radiative transfer simulations based on the temporally closest radiosonde and dropsonde profiles are used to estimate the fraction of direct solar radiation. For the simulations, the one-dimensional (1D) plane-parallel radiative
transfer solver DIScrete ORdinaTe DISORT 2.0 embedded in the library for radiative transfer is applied (libRadtran, Emde et al., 2016; Stamnes et al., 2000). In cloudy conditions, no correction can be applied. Therefore, final BACARDI data includes both uncorrected $F_{\mathrm{sol}}^{\downarrow}$ to be used for cloudy conditions and a corrected product to be used in cloud-free conditions.

For the correction, the offset angles of BACARDI, $\Theta_0$ for the roll and $\Phi_0$ for the pitch angle, characterizing the relative alignment of the radiometer with respect to the inertial navigation system of HALO are determined from measurements during
test flights. In cloud-free conditions, flight sections in different flight directions are compared to simulations of the theoretical





downward solar irradiances. By minimizing the differences between corrected and simulated $F_{\mathrm{sol}}^{\downarrow}$, the best fitting pair of $\Theta_0$ and $\Phi_0$ are derived. For the installation of BACARDI during EUREC$^4$A, two test flights are analyzed, one performed before the campaign in Oberpfaffenhofen, Germany, and one during the campaign in Barbados. In both cases, $\Theta_0 = +0.3°$ and $\Phi_0 = +2.5°$ are obtained, so it can be assumed that the offset angles are stable once BACARDI is installed on HALO. To

account for the limitations of the attitude correction, the downward solar irradiance is filtered before publishing the data set. Data are assumed to be valid, when the attitude correction factors are less than 25 %. For larger correction factors, roll and pitch angles need to be smaller than 5°. The excluded data corresponds to turns with large roll angles or conditions with low Sun.

## 5   Thermal offset correction

As discussed in Section 2.3 and indicated by the sensor temperatures shown in Fig. 3, thermal offsets need to be considered if the radiometers are exposed to fast temperature changes. $\Delta F$ is expected to be proportional to the time derivative of the sensor reference temperature (Eq. 17). To quantify and finally correct this effect for BACARDI as operated on HALO, an exemplary night flight was performed on 15 May 2019. The flight represents a typical HALO ascent and descent profile including a few level steps before reaching a maximum height of about 13 km. The static air temperature varied between -55 °C and +20 °C, and

the highest flight level was clearly within the stratosphere. Take-off time was more than 1.5 h after sunset, therefore ensuring that no solar radiation was present.

During the night flight, it is assumed that the solar irradiances measured by the pyranometer are zero, $F_{\mathrm{sol}} = 0\,\mathrm{W\,m^{-2}}$. Thus, deviations from zero are used to quantify the thermal offset $\Delta F$. According to the theory, $\Delta F$ mainly depend on the derivative of the sensor reference temperature $\partial T_{\mathrm{ref}}/\partial t$. For pyranometer measurements from the night flight, this relation is

shown in Fig. 5. Since the calculation of $\partial T_{\mathrm{ref}}/\partial t$ amplifies the measurement noise, the signal is smoothed with a 10 s running mean filter before and after applying the derivative function. Through this treatment, no significant additional noise is added to the thermopile measurement when applying the thermal correction to the raw data. To remove long term trends of the ambient temperature and instrument performance, the data are additionally detrended with a high-pass filter. For the pyranometers, two averaging times are applied in the high-pass filter, 100 s displaying only very fast sensor responses and 1000 s, which also

includes slower adjustments of the thermal equilibrium. Both filters result in an almost identical trend that indicates that the pyranometers respond similarly to fast and slow temperature changes.

To quantify the thermal offset correction, two different fit approaches are selected. A simple linear fit (see Fig 5), which neglects the thermal dome effect, provides the correction coefficient $\beta$ in units of $\mathrm{W\,m^{-2}\,K^{-1}\,s}$. A more complex multi-variable fit (not shown here) including the absolute value of the sensor reference temperature $T_{\mathrm{ref}}$ following Eq. (17) is applied

but did not show significant improvement as a correction. This result shows that the dynamic dome effect can hardly be discriminated from the thermal offset of the thermopile, and the simple linear fit sufficiently corrects for $\Delta F$. All correction coefficients derived for the upper and lower pyranometer are listed in Table 1. For detrending the data with the 100 s high-pass filter, the coefficients of the upper pyranometer, $\beta = 235\,\mathrm{W\,m^{-2}\,K^{-1}\,s}$ and lower pyranometer, $\beta = 439\,\mathrm{W\,m^{-2}\,K^{-1}\,s}$,



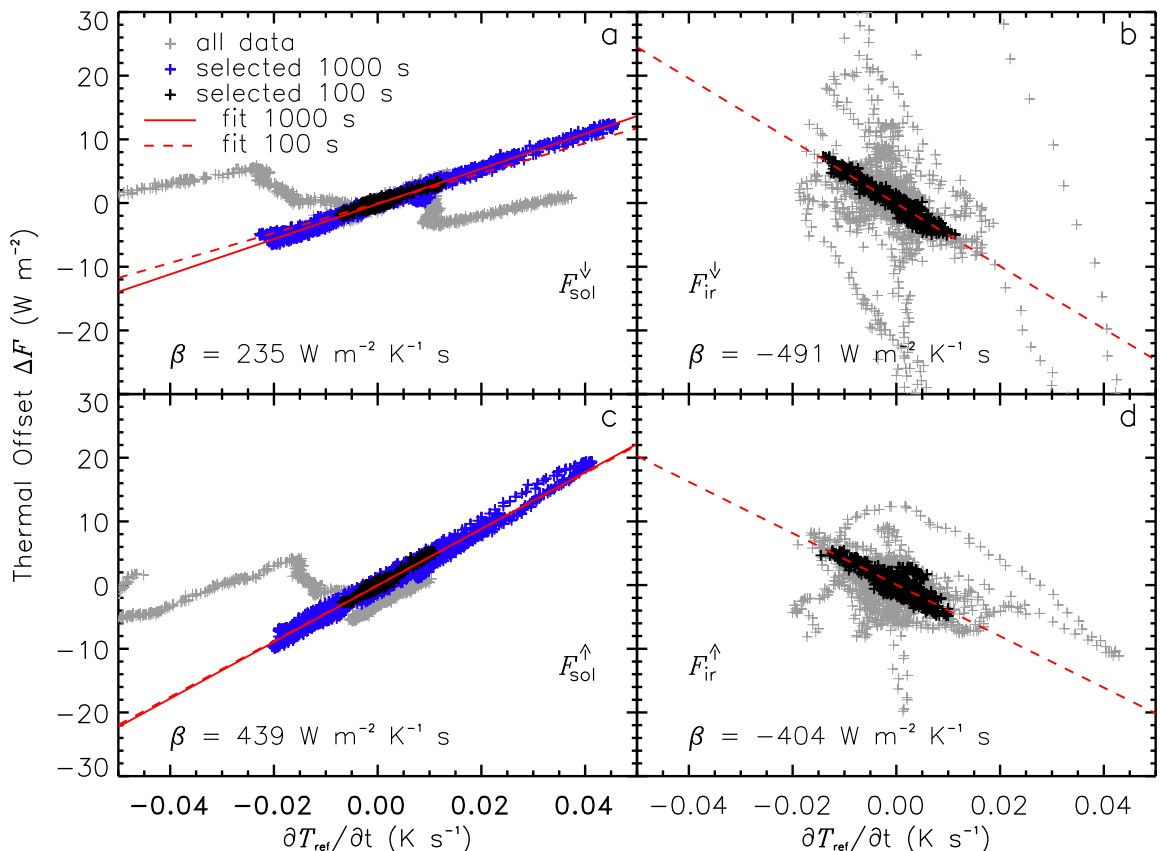

**Figure 5.** Thermal offset $\Delta F$ in dependence of the rate of temperature change for both pyranometers, downward solar irradiance (**a**) and upward solar irradiance (**c**), and both pyrgeometers, downward thermal-infrared irradiance (**b**) and upward thermal-infrared irradiance (**d**). For the selected and detrended data (1000 s and 100 s high-pass filter), linear regressions and the thermal offset correction coefficient $\beta$ (only for 100 s high-pass filter) are added.

significantly differ by almost a factor of two. This indicates that the lower radiometer is more strongly exposed to the air flow

(slight negative pitch angle of HALO) and is affected by a stronger thermal offset. Therefore, the coefficients reported here for BACARDI operated on HALO can not reliably be transferred to other broadband radiometers on other research aircraft.

For the thermal-infrared irradiance measured by the pyrgeometers, the assumption of $F_{\mathrm{ir}} = 0\,\mathrm{W\,m^{-2}}$ does not apply. On time scales of several minutes, $F_{\mathrm{ir}}$ also varies with changing atmospheric conditions and altitude and can not be assumed to be constant. Therefore, only the detrending with the 100 s high-pass filter is applied. Additionally, only selected flight seg-

ments are used to determine $\Delta F$. These sections are characterized by small variations in the thermal-infrared irradiance that match strong variations in temperature. The selected data are shown in Fig. 5b and 5d for both pyrgeometers. The remaining fluctuations of the pyrgeometers show an excellent correlation with $\partial T_{\mathrm{ref}}/\partial t$, which is inverse to the correlation of the

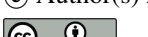



**Table 1.** Coefficients for thermal offset correction $\beta$ of the individual radiometers of BACARDI. To detrend the data, two high-pass filters (100 s and 1000 s) are applied for the pyranometer.

| Radiometer | $\beta$ (W m$^{-2}$ K$^{-1}$ s) | |
| --- | --- | --- |
| | for 100 s | for 1000 s |
| $F_{sol}^{\downarrow}$ | 235 | 276 |
| $F_{sol}^{\uparrow}$ | 439 | 444 |
| $F_{ir}^{\downarrow}$ | -491 | – |
| $F_{ir}^{\uparrow}$ | -404 | – |

pyranometers. The thermal offset correction coefficient amounts to $\beta = -491\,\mathrm{W\,m^{-2}\,K^{-1}\,s}$ for the upper pyrgeometer and $\beta = -404\,\mathrm{W\,m^{-2}\,K^{-1}\,s}$ for the lower pyrgeometer. Compared to the CMP22 pyranometers, $\beta$ of both CGR4 pyrgeometers

show only a small difference. This might be a consequence of the construction of the BACARDI sensor mounting, where the CGR4 are placed in front of CMP22 with respect to the flight direction and are therefore ventilated more effectively.

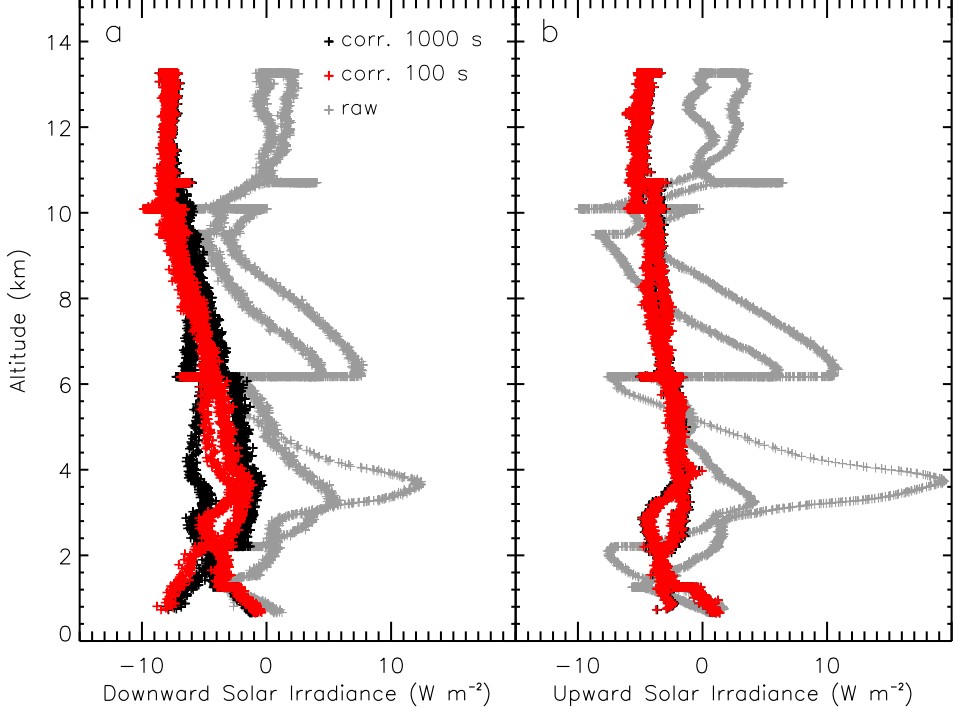

**Figure 6.** Profile of downward (**a**) and upward (**b**) solar irradiance during a night flight on 15 May 2019. Irradiances without (gray) and with thermal offset correction using the fast (100 s high-pass filter, red) and slow (1000 s high-pass filter, black) response fit are shown.


Applying these parametrizations, all irradiances measured during the night flight were corrected. In Fig. 6, the upward and downward solar irradiance are compared to the uncorrected measurements for the entire flight. The data are presented as vertical profiles to compare the ascent and descent, which should agree after correction in the absence of any solar radiation.

It is obvious that the uncorrected data show a similar pattern of fluctuations for upward and downward irradiance originating from the thermal effects. The thermal offset correction reduces this thermal error in both pyranometers from up to $20\,\mathrm{W\,m^{-2}}$ to a few $\mathrm{W\,m^{-2}}$. For the downward irradiance, the best agreement is found using the $100\,\mathrm{s}$ high-pass filter, while for the upward irradiance both filter options agree. The remaining bias to $F_{\mathrm{sol}} = 0\,\mathrm{W\,m^{-2}}$ is caused by different uncertainties such as the radiometric calibration of the pyranometer.

## 350  6   Measurement examples

Measurements of BACARDI during the EUREC$^4$A field campaign (Stevens et al., 2021) are used to demonstrate how the applied corrections affect typical analysis of broadband radiation measurements.

### 6.1   Irradiance and heating rate profile

The thermal offset correction is most relevant when the temperature environment changes rapidly, such as during ascents and

descents. Also, the aircraft flight velocity and the air density change the air flow around the sensors and control the adjustment of the thermal equilibrium. Figure 7 shows corrected and uncorrected profiles of all four irradiance components for an ascent up to $10\,\mathrm{km}$ altitude measured right after the start of the research flight on 7 February 2020 (Flight ID HALO-0207). To interpret this profile, it needs to be considered that during such an ascent, HALO also covers a horizontal distance of about $200\,\mathrm{km}$ during which the atmospheric conditions may change. However, to estimate the effect of the thermal offset correction

on the measurements, this case is well suited. Flight sections that do not comply at all with the required conditions, e.g., flight maneuvers of HALO, have been removed from the corrected data.

The ascent is characterized by an apparent cloud layer with cloud top at about $2\,\mathrm{km}$ as indicated by the increase of $F_{\mathrm{sol}}^{\downarrow}$ and the decrease of $F_{\mathrm{ir}}^{\downarrow}$ at this altitude. Above this cloud, cloud-free conditions above the aircraft prevail. The upward irradiances, solar and thermal-infrared, are both affected by the changing cloud situations below HALO. The general increase of reflected

solar radiation above the low-level cloud layer is covered by $F_{\mathrm{sol}}^{\uparrow}$, while $F_{\mathrm{ir}}^{\uparrow}$ drops only for a limited period. Afterwards, the low-level cloud layer likely became thinner along the flight track resulting in a cloud top temperature that is more similar to the surface temperature.

This general pattern is shown by both uncorrected and corrected data. Differences, as provided in Fig. 7c and 7f, increase with altitude. In general, the uncorrected data underestimates the solar irradiance and overestimates the thermal-infrared irra-

diance. This is due to the inverse correlation of temperature change and thermal offset for the CGR4 and CMP22 radiometer as discussed in Section 5. While both CGR4 pyrgeometers show an almost synchronized pattern (almost identical $\beta$), the thermal offset correction differs for both pyranometers (higher $\beta$ for $F_{\mathrm{sol}}^{\uparrow}$). Therefore, the upward solar irradiance is more affected than





the downward irradiance. As the thermal offset is independent of the absolute magnitude of the irradiance, this behavior might also be valid for other conditions with higher surface reflectivity or the presence of more reflective clouds.

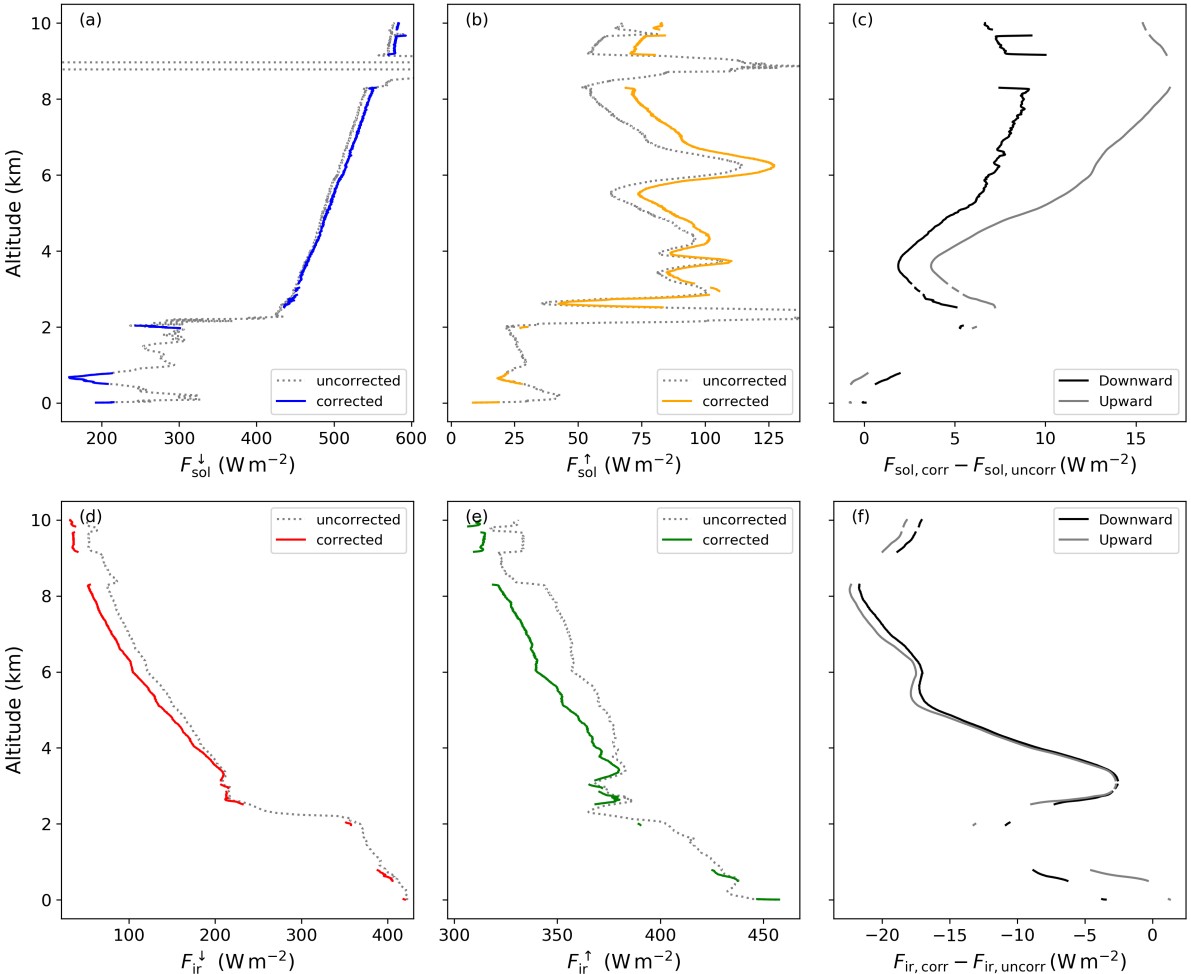

**Figure 7.** Vertical profiles of downward solar (**a**), upward solar (**b**), downward thermal-infrared (**d**), and upward thermal-infrared (**e**) irradiance given by the solid lines, respectively. The dotted gray lines indicate the corresponding profiles prior to the thermal correction. The two panels **c** and **f** show the absolute differences between the corrected and uncorrected profiles, downward (gray) and upward (black), for both the solar and thermal-infrared irradiance.

Profiles of broadband solar and thermal-infrared irradiance are often used to study and quantify the impact of clouds or aerosol layers on atmospheric heating rates. To calculate atmospheric heating rate profiles, the upward and downward irradiances are combined into the net irradiance $F_{\mathrm{net}}$, independently for both spectral ranges:

$$F_{\mathrm{net}} = F^{\downarrow} - F^{\uparrow}. \tag{18}$$



For the measurement case shown above, the net irradiance profiles for corrected and uncorrected data and their differences

are shown in Fig. 8. Differences between corrected and uncorrected data are below $8 \, \mathrm{W \, m^{-2}}$ for the solar irradiance and

below $5 \, \mathrm{W \, m^{-2}}$ for the thermal-infrared irradiance. As the upward and downward radiometers are almost equally affected

by the temperature change during the ascent, the thermal correction mostly cancels out for $F_{\mathrm{net}}$. Only the upper and lower

pyranometer show slight differences. This implies that also the uncorrected irradiances can be used to estimate $F_{\mathrm{net}}$. The

thermal offset correction becomes only relevant for the profiles of $F_{\mathrm{net,sol}}$ at higher altitudes, where temperature changes are

quicker. The net thermal-infrared irradiance, $F_{\mathrm{net,ir}}$, significantly differs only for low altitudes below the cloud layer.

Consequently, also the atmospheric heating rates, defined as the vertical change of net irradiance,

$$\frac{\partial T}{\partial t} = \frac{1}{\rho \cdot c_{\mathrm{p}}} \cdot \frac{\partial F_{\mathrm{net}}}{\partial z}, \tag{19}$$

show only a minor impact of the radiometer thermal offsets. In Eq. 19 $\rho$ represents the air density and $c_{\mathrm{p}}$ the heat capacity of

the air. From the example profile, heating rates are calculated for a $50 \, \mathrm{m}$ layers thickness showing the strongest heating rates of

down to $-4 \, \mathrm{K \, h^{-1}}$ at the top of the low-level cloud layer. However, the differences between corrected and uncorrected data are

less then $\pm 0.2 \, \mathrm{K \, h^{-1}}$ for the entire profile. These results demonstrate that for this specific application of BACARDI, which is

based on differences of upper and lower broadband radiometer measurements, a thermal offset correction could be neglected.

This might not be valid if the mounting position of BACARDI on HALO changes for other missions, which deviate from the

instrument configurations presented by Stevens et al. (2021).

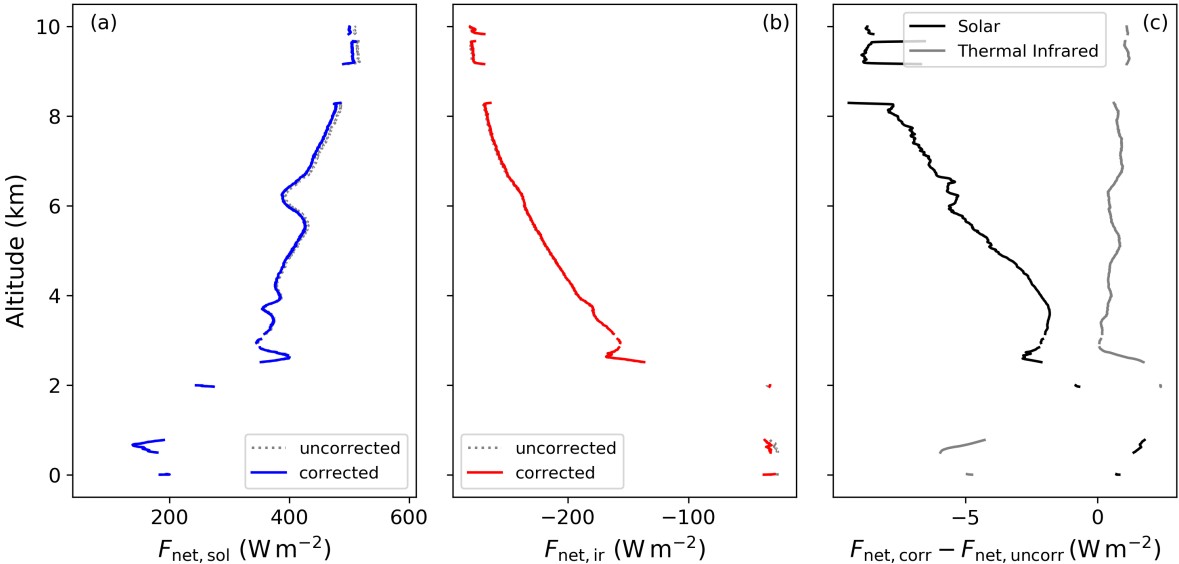

**Figure 8.** Vertical profiles of the solar (**a**) and thermal-infrared (**b**) net irradiance based on the thermal-corrected measurements. The gray dotted lines indicate the corresponding profiles prior to the application of the thermal correction. Panel **c** illustrates the difference between the corrected and uncorrected $F_{\mathrm{net}}$ for the solar and the thermal-infrared ranges, respectively.

## 6.2 Measurements during horizontal, circular flight pattern

During EUREC[4]A, HALO frequently flew a circular flight pattern that aimed to quantify the large-scale vertical motion, an eminent parameter characterizing the dynamic state of the atmosphere (Bony et al., 2017). The typical circle had a diameter of roughly $220\,\mathrm{km}$, which correspond to a permanent roll angle $\Phi$ of HALO between $2°$ and $3°$. Therefore, the correction of the downward solar irradiance $F_{\mathrm{sol}}^{\downarrow}$ for horizontal misalignment, as described in Section 4.3, becomes more important. At the same time, a circular flight pattern provides observations over the full range of relative solar azimuth angles and is thus an ideal test bed for evaluating the performance of the solar irradiance measurements.

The accuracy of the attitude correction is tested against radiative transfer simulations. The HALO flights of EUREC[4]A have mostly been performed at flight altitudes above $10\,\mathrm{km}$ and under often cloud-free conditions above HALO, for which simulations of $F_{\mathrm{sol}}^{\downarrow}$ are reliable and can serve as a benchmark. The simulations have been performed along the HALO track, considering the time of day, the geographical position, and flight altitude of HALO with a temporal resolution of at least $30\,\mathrm{s}$. The radiative transfer solver DISORT 2.0 and the lowtran parameterization of molecular absorption embedded in libRadtran are applied (Emde et al., 2016; Stamnes et al., 2000; Ricchiazzi and Gautier, 1998). In the simulations, the cloud-free atmosphere is defined by merged temperature and humidity profiles from the Barbados Cloud Observatory radiosondes (BCO, Stevens et al., 2016; Stephan et al., 2021), and the frequent dropsonde measurements from HALO (George, 2021). The absorption by ozone, which becomes relevant for the typical flight altitude of $10\,\mathrm{km}$, is determined by satellite estimates of the atmospheric ozone column. The sea surface albedo is parameterized on the basis of Cox and Munk (1954), using the $10\,\mathrm{m}$ wind speed obtained from the lower most wind speed value of the HALO dropsondes.

Figure 9 compares downward and upward solar irradiance, $F_{\mathrm{sol}}^{\downarrow}$ and $F_{\mathrm{sol}}^{\uparrow}$, measured by BACARDI during the entire flight of 7 February 2020 (Flight-ID HALO-0207) with along-track simulations for cloud-free conditions. To illustrate the effect of the attitude correction for the downward irradiance, data with (black line) and without attitude correction (blue line) are plotted. Superimposed to the a diurnal cycle, the uncorrected $F_{\mathrm{sol}}^{\downarrow}$ shows oscillations with different frequency. The slow oscillations (between 0.5-1 hour) are associated with the circular flight pattern and caused by a combination of a permanent roll angle of about $3°$ and changes of latitude (solar zenith angle). Oscillations with higher frequencies between $0.5\,\mathrm{Hz}$ and $1\,\mathrm{Hz}$, e.g., most obvious between 17:00–19:30 UTC, result from variations of the roll and pitch angle due to turbulence and the aircraft autopilot. The post-correction of $F_{\mathrm{sol}}^{\downarrow}$ does remove most of these fast and slow oscillations. This confirms that the roll and pitch angle offsets are determined with sufficient accuracy and that the sensor response time of BACARDI is corrected, so that the oscillations are synchronized in time with the aircraft attitude. Subsequently, the remaining slow oscillations are in phase with the simulations and are only caused by the changes of the local solar zenith angle (latitude).

A statistically more robust comparison of measured and simulated $F_{\mathrm{sol}}^{\downarrow}$ is performed merging 12 EUREC[4] flights and filtering the data for cloud-free conditions and reliable attitude corrections. Data are assumed to be valid when the attitude correction factors are less than $25\,\%$. For larger correction factors, roll and pitch angles need to be smaller than $5°$. A one-to-one comparison is shown in Fig. 10a. From the total number of almost 3 million individual measurement samples, $97.6\,\%$ agree with the simulations within an uncertainty range of less than $5\,\%$. This indicates that the general performance of BACARDI including



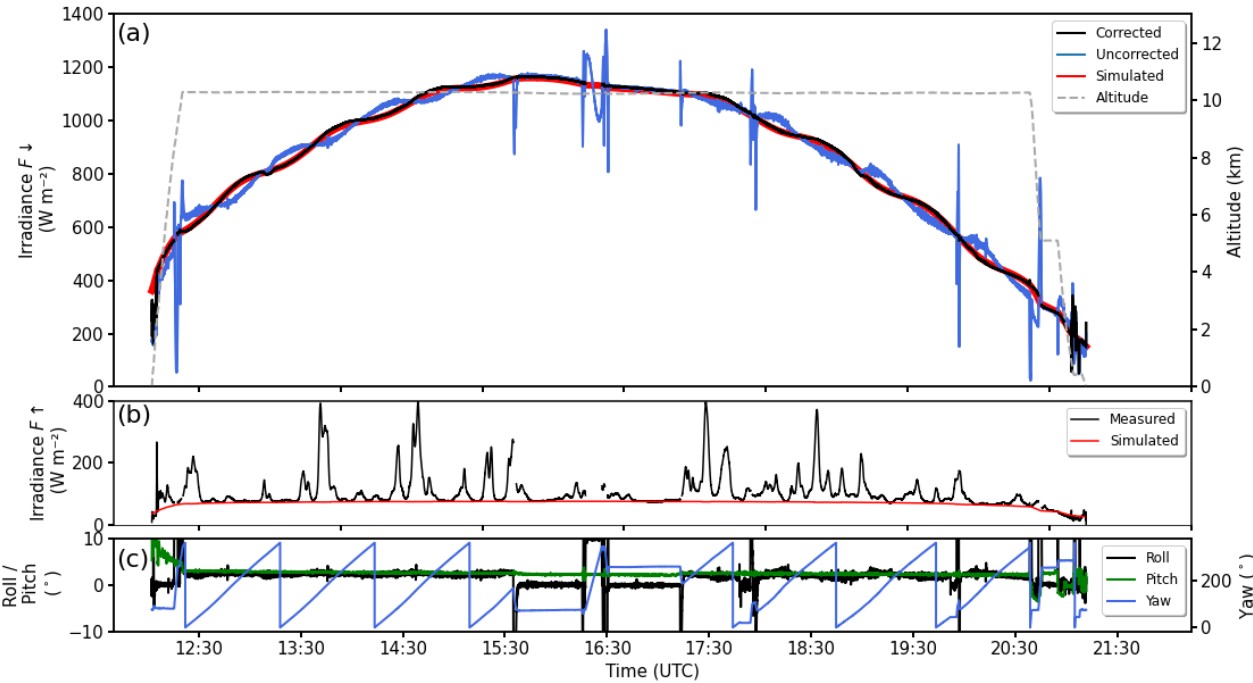

**Figure 9.** Time series of downward solar irradiance $F_{sol}^{\downarrow}$ (**a**) and upward solar irradiance $F_{sol}^{\uparrow}$ (**b**) measured by BACARDI on 07 February 2020 (Flight-ID HALO-0207). For the downward component, data with and without attitude correction are given (label corrected and uncorrected). For comparison, along-track simulations of $F_{sol}^{\downarrow}$ and $F_{sol}^{\uparrow}$ for cloud-free conditions are shown. The flight altitude is presented in panel (**a**), the aircraft attitude is given by the roll, pitch, and yaw angles in panel (**c**).

the thermal and attitude correction is stable over the entire campaign. A linear regression of all reliable filtered data shows

only a slight deviation from the 1:1 slope with a correlation coefficient of 0.999. The absolute differences are limited by the applied filter but illustrate that for high solar irradiances, the outliers of the measurements tend to be lower than the simulations. This might be caused by a false detection of clouds above the aircraft, which are not considered in the cloud-free simulations. For low values of $F_{sol}^{\downarrow}$ the measurements are slightly overestimated. These measurements correspond to conditions of low solar zenith angle, when the attitude correction becomes more critical. At the same time, the angular response of the CMP22

pyranometer is known to slightly deviate from an ideal cosine response at high solar zenith angles.

      Making use of the circular flight pattern, a potential asymmetrical cosine response of the pyranometer inlet is investigated in Fig. 10b. The ratio of the corrected observations and simulations is analyzed as a function of solar zenith angle $\theta_0$ and relative heading of HALO with respect to the solar azimuth. Only a subset of seven flights are used. Other flights are excluded because they either did not contain the circle flight pattern or show evidence of contamination by higher clouds like cirrus. Up

to solar zenith angles of approximately $75°$, the observed $F_{sol}^{\downarrow}$ is within $5\,\%$ of the simulated values. The good agreement of





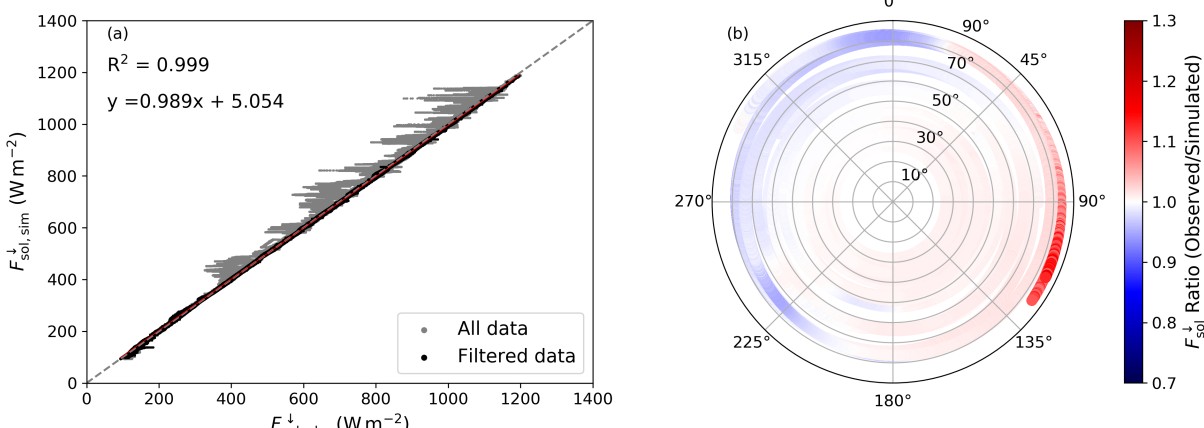

**Figure 10. (a)** Comparison of simulated and observed downward solar irradiance $F_{sol}^{\downarrow}$ of 12 EUREC[4]A flights. The gray dots include all data points, while the black dots are filtered for cloud-free conditions and a reliable attitude correction. **(b)** The ratio of the observed to the simulated $F_{sol}^{\downarrow}$ (color code) as a function of solar zenith angle (radial axis) and the heading angle of HALO relative to the solar azimuth angle $\theta_0$ for the filtered subset of panel a.

the majority of the data points is regarded as an indicator that the attitude correction is independent of the flight direction over a wider range of illumination conditions ($0 - 1200 \, \mathrm{W \, m^{-2}}$) and solar zenith angles ($0 - 75°$).

For solar zenith angles larger than $75°$ a slight directional dependence, relative position of the Sun with respect to the orientation of BACARDI (HALO), is obvious. $F_{sol}^{\downarrow}$ is overestimated by BACARDI between $30 - 210°$ relative solar azimuth

and underestimated if the Sun is in the opposite directions. These effects may result for different reasons, which can not be disentangled here. It might indicate slightly incorrect offset angles determined for the attitude correction, an azimuthal dependence of the cosine response of the pyranometer, or reflection by the aircraft fuselage and the tail-plane fin at high zenith angles. Therefore, it is advisable to use the data at $\theta_0 > 75°$ with some amount of caution.

The upward solar irradiance can not be corrected for the aircraft attitude but at the same time is characterized by a more

isotropic radiation field compared to the direct solar radiation. The time series of $F_{sol}^{\uparrow}$ shown in Fig. 9b indicates that the flight track covers an area with a generally low cloud cover and some patches of low-level stratiform clouds. The cloud-free areas correspond to the low values of $F_{sol}^{\uparrow}$, which form a baseline at about $80 \, \mathrm{W \, m^{-2}}$. Only these cloud-free measurements can be compared to radiative transfer simulations (red line). The measurements match the simulated baseline and also follow their slight diurnal change with higher values observed at solar noon. The agreement of observed and simulated $F_{sol}^{\uparrow}$ indicates that

the measurements in conditions like EUREC[4]A are reliable, even without any attitude correction. For observations over higher reflecting surfaces like sea ice, this needs to be confirmed.





## 7 Conclusions

A new radiometer package, the Broadband AirCrAft RaDiometer Instrumentation (BACARDI) for the HALO research aircraft, is introduced and characterized. BACARDI comprises two sets of upward and downward looking broadband radiometers covering the solar and thermal-infrared spectral ranges. The operation of broadband pyranometers and pyrgeometers on airborne platforms in a challenging dynamic environment is investigated in this paper. Especially for a fast and high-flying aircraft such as HALO, where the environmental conditions such as air temperature and density can change rapidly, a minimization of these effects by constructional measures and a correction of the data is mandatory. Three basic corrections are applied to the measurements of BACARDI:

- The post-processing of BACARDI measurements accounts for changes of the sensor thermopile sensitivity. Due to the large range of environmental temperatures under which HALO operates (from the surface to the lower stratosphere), this correction amounts to about $5\,\mathrm{W\,m^{-2}}$ for the pyranometers, while the pyrgeometers sensitivity is more stable.

- The corrections of the sensor response time makes use of the $10\,\mathrm{Hz}$ sampling frequency and accounts for the fast change of irradiance, e.g., in case of crossing cloud or sea ice edges. The deconvolution method by Ehrlich and Wendisch (2015) with a response time of $1.2\,\mathrm{s}$ and $3.3\,\mathrm{s}$ for the pyranometers and pyrgeometers, respectively, is applied to reconstruct the high-frequency changes of irradiance.

- For the rather smooth changes of the HALO attitude (roll and pitch angle), the common correction method by Bannehr and Schwiesow (1993) is successfully applied to the downward solar irradiance as evaluated during circular flight pattern.

It is shown, that known thermal effects occur for BACARDI, when the sensor and dome temperatures do not change simultaneously, such as during ascents and descents into other temperature regimes. To correct for these thermal offsets, a new method is introduced. Historically, such effects were monitored and corrected with additional measurements of the dome temperature. The approach presented here is based on a simple parameterization that combines the dynamic dome effect and the thermal offset of the thermopile and, therefore, does not require measurements of the dome temperature. For the radiometers of BACARDI, the thermal offsets are found to correlate with the rate of change of the sensor temperature, which is expected from theory (see Eq. 17, Sec. 5). Using the sensor temperature as the proxy to determine the thermal offsets makes the post-processing straight forward as the sensor temperature is measured by the radiometers by default.

The parameterization of the thermal offset of BACARDI is derived from an exemplary calibration flight in nighttime conditions, where the pyranometer measurement can be assumed to be zero. For the pyrgeometers, selected flight sections with strong temperature changes are analyzed. The magnitude of the correction coefficients of the individual radiometer are in the range of $200\text{--}600\,\mathrm{W\,m^{-2}\,K^{-1}\,s}$ and depend on the position of the radiometer and the aircraft angle of attack. As the radiometer position and environmental conditions might change between HALO missions, the coefficients should be determined regularly. It also has to be noted that the coefficients reported for BACARDI operated on HALO can not be transferred to other broadband radiometers on other research aircraft.



The performance of BACARDI was evaluated by measurement examples from the EUREC[4]A field campaign (Stevens et al.,
2021). BACARDI was implemented on HALO for the first time during EUREC[4]A. The system extends the existing suite of
active and passive remote sensing instruments on HALO, which lacked instrumentation to observe the solar and thermal-
infrared radiative energy budget. BACARDI measurements during an ascent up to $10\,\mathrm{km}$ altitude demonstrate how strong the
new thermal offset affects the single irradiance components in fast changing environmental conditions. In general, without
thermal offset correction, the solar irradiance is underestimated while the thermal-infrared irradiance is overestimated by up to
$20\,\mathrm{W\,m^{-2}}$. The exact offset correction depends on the mounting position of the radiometer and air flow around the aircraft but
is independent of the magnitude of irradiance.

It is shown that net irradiances and atmospheric heating rates calculated from the upward and downward irradiances are less
affected by the thermal effect. As upper and lower radiometers show a similar magnitude of the thermal offset, the thermal
effects cancel out to a large extent. In contrast to ascents and descents, for straight flight legs maintaining constant flight levels,
which are more typical for HALO observations, the temperature changes are small, and potential thermal offsets range below
$1\%$ of the broadband irradiances, which appears negligible. Nevertheless, temperature variations and sudden temperature
gradients can appear along constant height levels, e.g., at upper-level frontal systems or tropopause disturbances.

The circular flight pattern frequently performed during EUREC[4]A required special care for the attitude correction of $F_{\mathrm{sol}}^{\downarrow}$.
Comparisons of the measurements to cloud-free radiative transfer simulations indicate that remaining biases after applying
the attitude correction are significant only for solar zenith angles larger than $75°$, which were present during EUREC[4]A only
briefly during early or late flights.

The processed broadband irradiances measured by BACARDI during EUREC[4]A are published at the AERIS atmosphere
Data and Services Centre (Ehrlich et al., 2021). The data are used by (Luebke et al., 2022) to assess the cloud radiative forcing
with regard to the cloud life cycle and the cloud's temporal evolution, both of which are targets of EUREC[4]A.

*Data availability.*   Processed data of BACARDI are published at the AERIS atmosphere Data and Services Centre (Ehrlich et al., 2021). Raw
data can be obtained from the authors on request.

*Author contributions.*   **AE** and **MZ** equally contributed to the manuscript. **AE** compiled the manuscript, performed the basic corrections of
the BACARDI measurements and evaluated the thermal offset correction. **MZ** invented the concept of the thermal offset correction, which is
presented in this paper, and is responsible for the design BACARDI project and the integration into the aircraft. **AG**, **VN**, and **CM** supported
**MZ** and performed the thermal correction. **RM** and **TR** designed the instruments and provided the construction drawing. **BS** and **MW** were
responsible for the scientific guidance of the project and designed and coordinated the flight strategy. **AE**, **AL**, **KW** provided the scientific
background and the data analysis for the presented measurement examples.

*Competing interests.*   No competing interests are present.



*Acknowledgements.* We thank the Max Planck Institute for Meteorology, Hamburg, Germany, for the funding of the new radiometer system
and providing it to the HALO community. We are further grateful for funding of the project grant nos. 422897361 and 316500630 by the DFG
within the framework of the Priority Programme SPP 1294 to promote research with HALO. We gratefully acknowledge the funding by the
Deutsche Forschungsgemeinschaft (DFG, German Research Foundation) – project number 268020496 – TRR 172, within the Transregional
Collaborative Research Center "ArctiC Amplification: Climate Relevant Atmospheric and SurfaCe Processes, and Feedback Mechanisms
(AC)3". We thank the Max Planck Institute for Meteorology for designing and coordinating the EUREC$^4$A campaign and the German
Aerospace Center (Deutsches Luft und Raumfahrtzentrum, DLR) for campaign support.





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
