# Peer review of "A new airborne broadband radiometer system and an efficient method to correct dynamic thermal offsets"

_Atmospheric Measurement Techniques, 2022_

## Author Comment (AC1)

**Reply on Referee #1 Stefan Wacker (AMT-2022-259)**

André Ehrlich[1,★], Martin Zöger[2,★], Andreas Giez[2], Vladyslav Nenakhov[2], Christian Mallaun[2], Rolf Maser[3], Timo Röschenthaler[3], Anna E. Luebke[1], Kevin Wolf[1,a], Bjorn Stevens[4], and Manfred Wendisch[1]

[1]Leipzig Institute for Meteorology, Leipzig University , Germany
[2]German Aerospace Center, Flight Experiments, Oberpfaffenhofen, Germany
[3]enviscope GmbH, Frankfurt am Main, Germany
[4]*Max Planck* Institute for Meteorology, Hamburg, Germany
[a]now at: Institute *Pierre-Simon Laplace*, Sorbonne Université, Paris, France
[★]These authors contributed equally to this work.

**Correspondence:** A. Ehrlich
(a.ehrlich@uni-leipzig.de)

**1 Introduction**

The comments of the reviewer have been helpful to improve the manuscript. We are especially thankful for pointing at the comparison to radiative transfer simulations of the thermal-infrared downward irradiance and the missing literature. This provided some new aspects to quantify the performance of the radiometer systems. We think that this significantly increased the value of the manuscript for potential readers.

The detailed replies on the reviewers comments are given below. The reviewers comments are given in bold while our replies are written in regular roman letters. Citations from the revised manuscript are given as indented and italic text.

**Detailed Replies**

**Corrections methods and their evaluation are thoroughly described for the shortwave and longwave. In addition, a comparison of the observations with radiative transfer calculations for the downwelling and upwelling shortwave radiative fluxes for horizontal, circular flight patterns has been presented. Such a comparison would be also highly valuable for the (downwelling) longwave in order to estimate the reliability of the longwave observations and to study potential effects which may become relevant in the longwave on such flights. For instance, the fraction of the direct solar beam above the cut-on of a pyrgeometer, which is at about 4.5 $\mu$ m for a CGR4. This unintentionally observed portion of the direct solar beam depends on the water vapor content (and thus altitude) and the solar zenith angle and hence may exceed $5\,\mathrm{W\,m^{-2}}$ significantly on such flights (e.g., Marty, 2000). In addition, the dependency of the pyrgeometer sensitivity on the water vapor content, which is estimated to be about $5\,\mathrm{W\,m^{-2}}$ in cloudfree conditions, may also be considered in such applications (e.g., Nyeki et al.,2017).**

Thanks for this valuable suggestion! We did not simulated the thermal-infrared downward irradiance due to the higher sensitivity to the water vapor column above the aircraft and the distance to the radiosondes. That's why we did not expect to be able to

identify this bias. We now run the simulations of downward thermal-infrared irradiance for the EUREC4A measurements with the best estimate of the atmosphere profile and compared to the measurements. Indeed, a slight positive bias of up to $10\,\mathrm{W\,m^{-2}}$ is observed when the downward solar radiation is high. In the revised manuscript, we included this analysis and discussed the offset.

In the abstract we added:

> *Special materials, e.g., quartz glass, silicon, as well as filter coatings guarantee a relatively constant sensitivity of the instrument over the desired spectral range (Groebner et al., 2007). However, unshaded pyrgeometer may suffer from leakage effects when solar radiation is transmitted above the cut-on wavelength of the pyrgeometer dome interference filter (Marty 2000; Meloni et al., 2012).*

The new section reads:

[Figure]

**Figure 1.** Difference between measured and simulated downward thermal-infrared irradiance $\Delta F_{ir}^{\downarrow}$ for 12 EUREC[4]A flights filtered for cloud-free conditions and flight altitudes above $10\,\mathrm{km}$. The differences are plotted and fitted as function of the measured downward solar irradiance $F_{sol}^{\downarrow}$.

> *The HALO flights of EUREC[4]A have mostly been performed at flight altitudes above $10\,\mathrm{km}$ and under often cloud-free conditions above HALO, causing high values of downward solar irradiance. In such conditions the solar leakage of the pyrgeometer dome interference filter can produce an overestimation of the thermal-infrared irradiance (Philipona et al., 1995; Marty, 2000; Meloni et al., 2012). For cloud-free ground-based measurements, Meloni et al. (2012) identified an overstimation of up to $10\,\mathrm{W\,m^{-2}}$ depending on the amount of downward solar irradiance.*

*This bias was investigated for BACARDI using radiative transfer simulations of $F_{ir}^{\downarrow}$ which are reliable and can serve as a benchmark for measurement above 10 km and under cloud-free conditions. The simulations have been performed along the HALO track, considering the time of day, the geographical position, and flight altitude of HALO with a temporal resolution of at least 30 s. The radiative transfer solver DISORT 2.0 and the lowtran parameterization of molecular absorption embedded in libRadtran are applied (Emde et al., 2016; Stamnes et al., 2000; Ricchiazzi and Gautier, 1998). In the simulations, the cloud-free atmosphere is defined by merged temperature and humidity profiles from the Barbados Cloud Observatory radiosondes (BCO, Stevens et al., 2016; Stephan et al., 2021), and the frequent dropsonde measurements from HALO (George, 2021).*

*Filtered for cloud-free condition, Figure 9 shows the difference between measured and simulated $F_{ir}^{\downarrow}$ as function of the measured downward solar irradiance $F_{sol}^{\downarrow}$. The data indicates a trend to overestimate $F_{ir}^{\downarrow}$ for increasing $F_{sol}^{\downarrow}$. For values of $F_{sol}^{\downarrow}$ above 1000 W m$^{-2}$, typical for times around solar noon, the bias ranges up to 10 W m$^{-2}$ comparable to the findings of Meloni et al. (2012). A linear regression suggest an increase of the bias by 1 W m$^{-2}$ for each 100 W m$^{-2}$ increase of $F_{sol}^{\downarrow}$. However, the data shows a large variability and the regression suggests a negative bias for the absence of solar radiation. This may be attributed to remaining uncertainties of the radiative transfer simulations and the pyrgeometer sensitivity due to changes of water vapor concentrations above HALO (Nyeki et al., 2017), a permanent biases of the radiometer calibration, and a static thermal offsets*

In the conclusions we added:

*In conditions with high $F_{sol}^{\downarrow}$, the pyrgeometer show a slight bias due to leakage of solar radiation above the cut-on wavelength of the CGR4 interferrence filter. This bias correlates with $F_{sol}^{\downarrow}$ and amounts up to 10 W m$^{-2}$ during solar noon.*

**Would it possible to calculate an uncertainty budget for the BACARDI package or at least to give a conclusive uncertainty estimate for the individual components of the observed radiative fluxes and the net radiation?**

As shown in the manuscript, the uncertainties the irradiances result from different sources and processes, which are partly compensated by the applied corrections. As the magnitude of the biases and the corrections depends on a number of parameters such as the change of ambient temperature, solar zenith angle, direct fraction of solar radiation, also the remaining uncertainties of the radiometer are not a constant value. It rather depends on the specific atmospheric conditions and flight pattern. In addition, such biases may be come more important for low values of irradiance while in contrast the uncertainty of the radiometer sensor sensitivity scales with the magnitude of the irradiance. Thus estimating a conclusive general uncertainty estimate is challenging. We rather leave it to the publication of specific applications of the data to sum up the individual sources of uncertainty discussed in the manuscript, e.g. for level flights in high altitude or profiles for deriving heating rates. To make it more clear, that despite all corrections, a major part of the uncertainty results from the radiometer sensitivities, we edited and added in the conclusion:

*... for straight flight legs maintaining constant flight levels, which are more typical for HALO observations, the temperature changes are small (below 5 K per hour), and potential dynamic thermal offsets range below 1 W m$^{-2}$ for all broadband irradiances, which appears negligible compared to the uncertainties of the sensor sensitivities (1 % for the CMP22 pyranometer and 4 % for the CGR4 pyrgeometer).*

**It is indicated that the thermal offset correction coefficient $\beta$ of the upper and lower pyrgeometer is more consistent compared to the upper and lower pyranometer due to the position of the pyrgeometers in front of the pyranometers with respect to the flight direction, which allows the pyrgeometers to be ventilated more effectively. Would it possible to place the pyranometer to the side of the pyrgeometers to further reduce thermal offsets or impedes the mounting system of BACARDI or limited space in the fuselage such a setup?**

This is a valid suggestion. However, the aperture plates on which the radiometers are mounted have limited space and allow only a in line alignment of both radiometers. In addition due to other constrains such as the limited number of central apertured and certification costs, the construction of BACARDI is fixed for the near future.

**Line 32: may use "... by radiometers, ... pyranometers ... pyrgeometers"**

Thanks! We now tried to consistently use plural throughout the manuscript.

**Line 55: may use "Actively stabilized pyranometers, ..."**

Thanks! We now tried to consistently use plural throughout the manuscript.

**Line 83: may use "The radiative energy budget of a broadband radiometer"**

Thanks! We now tried to consistently use plural throughout the manuscript.

**Lines 129/140: In my opinion, the sensitivity is normally given in units of V W$^{-1}$ m$^2$ (see line 189). Hence, the stated unit W m$^{-2}$V$^{-1}$ refers to the reciprocal of the sensitivity à may use "... adjusted reciprocal of the pyrgeometer/pyranometer sensitivity..."**

You are right, the sensitivity of a sensor is defined the inverse way. Our intention was to make the equations which refer to the irradiance easier to read. Using a term like "reciprocal of sensitivity" we think will produce even more confusion. So finally, we decided to change the equation to the proper definition of sensitivity in units of V (W m$^{-2}$)$^{-1}$.

**Line 157: delete one "the"**

Thanks!

**Line 313: "... depends..."**

Thanks!

**Fig. 5: Is there an indication for the cause of the "outliers" in Fig. 5 (grey points)? Are these the same datapoints for the shortwave and longwave? May give a short statement in the text.**

The gray points in the plot show all data from the night flight (1 1/2 hours). The colored crosses show data during ascents and descents that were used to the regression (1 hour for the pyranometers and four short 3-4 min sections for the pyrgeometers). The sections of the pyrgeometers were selected by hand, when a significant reaction of the thermophile signal occurred during ascents and descents. For the pyranometer the conditions were more stable. Only sections during start and landing needed to be removed from the regression. However, for the pyranometer, the slope at these data (gray symbols) branches is similar to the regression, only that for zero temperature change, still a bias would exist. Reasons for that we could not be identify. For the pyrgeometer the variabiltity of the removed data (gray symbols) illustrates the variability of thermal-infrared radiation due to changes of the atmosphere which are not fully removed by detrending the data. In the revised manuscript, we added some more explanation in the text and the figure caption:

> *Data measured shortly after start and before landing (gray symbols) were not used to determine the thermal offsets.*

> *Gray symbols show all data of the night flight on 15 May 2019 (about 1.5 hours)*

**Line 346/347: I would replace "… a few W m-2" by "…to values below $10\,\mathrm{W\,m^{-2}}$" (as in the abstract). For the downwelling shortwave flux, values are rather between 5 and $10\,\mathrm{W\,m^{-2}}$ above 6 km but also partly near the surface. Only in the upwelling shortwave and in the downwelling shortwave up to 6 km values are a few $\mathrm{W\,m^{-2}}$**

That is correct. We adjusted to the statement given in the abstract that the bias is reduced "to values below $10\,\mathrm{W\,m^{-2}}$.

**Line 416: Delete either "the" or "a"**

Thanks!

**Line 432: I do not understand the expression "… false detection of clouds above the aircraft…". May rephrase, e.g., "… caused by cloud contamination above the aircraft in the filtered dataset, …" or similar. Significant lower observed irradiances with respect to the calculated fluxes are either due to real clouds above the aircraft or a not properly corrected misalignment of the sensor.**

Thanks for pointing at this unprecise expression. We changed the sentence in the revised manuscript:

> *This might be caused by a remaining contamination of the filtered data by clouds above the aircraft, which are not considered in the cloud-free simulations.*

**Line 433: "... conditions of high solar zenith angles, ..."**

Thanks again!

**Lines 413-456: I got a bit confused here: Fig. 9a is presented and described in lines 413-423. Then Fig. 10a and 10b are described in lines 424-448. Finally, you go back again to Fig. 9b in lines 449-456. It might be easier for the reader, if the description of Fig. 9b (lines 449-456) was placed right after the description of Fig. 9a in line 423. However, you may have good reasons not to do it.**

When writing the manuscript, we also struggeled with the issue that downward and upward irradiance are shown in Fig. 9 but Fig. 10 additionally addresses the downward irradiance. Finally we decided to discuss first all analysis for the downward irradiance (including Fig. 10) and then coming back to the upward irradiance. We think not splitting up the discussion on the attitude correction of the downward irradiance is more important and keep the section as it is.

---

## Author Comment (AC2)

**Reply on Referee #2 (AMT-2022-259)**

André Ehrlich1,★, Martin Zöger2,★, Andreas Giez2, Vladyslav Nenakhov2, Christian Mallaun2, Rolf Maser3, Timo Röschenthaler3, Anna E. Luebke1, Kevin Wolf1,a, Bjorn Stevens4, and Manfred Wendisch1

1Leipzig Institute for Meteorology, Leipzig University, Germany 2German Aerospace Center, Flight Experiments, Oberpfaffenhofen, Germany 3enviscope GmbH, Frankfurt am Main, Germany 4*Max Planck* Institute for Meteorology, Hamburg, Germany anow at: Institute *Pierre-Simon Laplace*, Sorbonne Université, Paris, France \*These authors contributed equally to this work.

**Correspondence:** A. Ehrlich (a.ehrlich@uni-leipzig.de)

**1 Introduction**

The comments of the reviewer have been helpful to improve the manuscript. Especially the discussion on the mounting position gave us some new inspirations. And the differentiation of static and dynamic thermal offsets will hopefully make the manuscript easier to understand for the interested readers. The detailed replies on the reviewers comments are given below. The reviewers comments are given in bold while our replies are written in regular roman letters. Citations from the revised manuscript are given as indented and italic text.

**Detailed Replies**

The abstract states that the correction function "depends on the mounting position of the radiometer on HALO." Please clarify what is meant by mounting position, and how conclusive this statement should be as several statements were made in the paper related to the impacts of mounting position, some of which seemed fairly conclusive and others more as hypotheses. In section 3.3 (Figure 3), the larger differences in sensor temperatures between the pyranometers than pyrgeometers rather than the upwelling or downwelling instruments is explained as being a matter of "the internal sensor housing" which I took to mean inherent to the differences in the instrument construction rather than how they were mounted on the plane. In section 5, the coefficients of the upper and lower pyranometer are described to differ by a factor of 2, and this is attributed to differing airflow between the two systems given the slight tilt of the plane. Then later in section 5, the up and downlooking pyrgeometers had much similar beta values which was hypothesized to be because "the CGR4 sensors are placed in front of the CMP22s". While there is a difference between sensor temperature agreement and agreement in coefficients for the corrections for dynamic offsets, they are related. I am curious whether the explanations that the authors give for these factors that cause differences in CMP22's and CGR4's thermal responses

**(position relative to airflow in section 5, and internal sensor differences in section 3.3) are perhaps related and how important the relative ventilation of the radiometers is thought to be in comparison to sensor differences.**

Thank for this discussion! Some of our conclusions and hypothesises were not given precisely and even contradict. You are right, that it is a combination of mounting position and the inherent differences in construction and internal heat transfer between pyranometer and pyrgeometer.

There is an indication, that the front position of the CGR4 sensors leads to a faster response of the sensor temperature (Fig. 3) although we can not rule out, that this is a consequence of a different internal construction of both radiometer. A more solid radiometer body or a different position of the sensor thermometer will also result in differences of the temperature adjustment. Additionally, the thermal offset is a result of the difference of dome temperature (no measured) and sensor temperature, and the dome material. Therefore, the ventilation and response to changes of ambient temperature can not directly be linked to the thermal offset coefficients  $\beta$ . It might be, that the internal ventilation of upper and lower sensor package is similar but the dome temperature adjusts different depending on the position and angle of attack. This may explain the differences of  $\beta$  for the upper and lower CMP22. But also heat conduction from the cabin or fuselage may add to the differences.

Finally,  $\beta$  needs to be determined for each setup (radiometer and aircraft). It is natural to assume, that better ventilation will help to reduce thermal offsets. However, from our analysis we can not quantify to what extend the mounting order (CGR4 in front of CMP22 or vice versa) impacts the thermal offsets. In the revised manuscript we did some corrections to make the reader sensitive to the combination of all potential causes:

Abstract: ... The parameterization provides a linear correction function (200–500 W m-2 K-1 s), that depends on the radiometer type and the mounting position of the radiometer on HALO.

Section 3.3: This indicates that temperature adjustments are rather a matter of the internal sensor housing of CGR4 and CMP22, their internal heat transfer, and the mounting order (CGR4 mounted in front of CMP22). The ventilation within the fairing is similar in upper and lower sensor package.

Section 5: This might be a consequence of the less exposed domes of the CGR4 compared to the CMP22 in combination with the more efficient ventilation of the CGR4 inside the BACARDI sensor mounting where the CGR4 is placed in front of CMP22 with respect to the flight direction.

Conclusion: The exact offset correction depends on radiometer type, the mounting position of the radiometer, and the air flow around the aircraft but is independent of the magnitude of irradiance.

I had a couple of questions about the practicality of the attitude correction method used. It seems to be sufficiently accurate, and the HALO aircraft remarkably level in most flights, so these are minor concerns that the authors shouldn't need to address for the publication of this paper. But I still found myself curious about a few practical details. The authors state that only the direct beam should be corrected for, so this correction should only be applied to clear sky conditions. I agree with that, however, I didn't understand how the data was determined to be clear or cloudy. In the test case, the profiles could be determined to be clear fairly easily by visual inspection, though I would imagine this would be a harder job for a full field campaign. Was this correction run at all times and it left to users to determine whether to use the corrected or uncorrected data or is some kind of determination made for a best estimate value? Also, as the correction was based on radiative transfer calculations using atmospheric profiles from drop sondes or radiosonde launches—I was curious whether these will always be available for all campaigns where the HALO flies?

You are right, the attitude correction of the solar downward irradiance can only by applied to the direct radiation for which geometry is known. The decision if and when, with what direct fraction to correct the measurements depends a lot on the application of the data and can be done with different degrees of accuracy. A general procedure for a full campaign is challenging. Our approach is the following: published BACARDI data is provided for both conditions: I) cloud-free (correction applied based on the diffuse fraction obtained from simulation of cloud-free conditions. II) cloudy (no attitude correction applied). In addition, the published data set includes the simulated downward irradiance assuming cloud-free conditions and a basic cloud-mask. The decision on cloud-free or cloudy is done by a combination of comparing measurements to simulations of the expected cloud-free irradiance and by analysing the variability of the downward solar irradiance within a 20 s running window. The variability threshold accounts for cirrus, where the irradiance may even exceed cloud-free conditions due to enhanced scattering. It is assumed, that clouds, in particular cirrus, lead to an enhanced variability. It is planned to provide the same data sets also for upcoming BACARDI measurements.

We hope this additional data provides sufficient information to make use of the data for the most applications. In the revised manuscript, we added the following information:

A basic cloud mask, that is based on a comparison with the expected cloud-free irradiance and the identification of enhanced variability of the downward solar irradiance within a 20 s running window, is provided int he published data set.

My primary concern with the methodology is that it wasn't clear to me in the text which thermal offset corrections are applied to the data (that derived in section 2.3 only, or also a correction derived in section 2.2). In Figure 6, after the dynamic linear-fit correction has been applied, there are still negative biases in the downwelling solar irradiance of  $5 \cdot 10 \text{ W m}^{-2}$  at night. I don't see an adequate explanation for what this bias is. The reason given in lines 348-349, "caused by different uncertainties such as the radiometric calibration of the pyranometer", seem quite hand-wavey and not satisfying to me compared to the careful work done elsewhere in deriving the corrections. A calibration error is multiplicative so shouldn't give a bias at night. It seems more likely to me from the shape of that bias (larger with higher altitudes) that it is in fact related to a thermal offset (like that derived in section 2.2) that isn't corrected for using the "beta" linear fit. The author's state in lines 325-326 that a more complex multi-variate fit including Tref doesn't improve the correction, and conclude that therefore the dynamic dome effect can't be discriminated from the thermal offset. But they don't show those results, and I still can't help but think that the postcorrection results in Figure 6 look like they are still impacted by an equilibrium thermal offset. Also, Figure 4 shows downwelling SW offset corrections even in level flights when the temperature doesn't appear to be changing significantly, which implies that the static offset is taken into account in some way. So it was unclear to me whether the dynamic offset (beta) correction was applied to this data or a static offset as derived in section 2.2.

Thanks for making this comment! We need to be more precise in our description of the processing.

What we can and do correct is the dynamic thermal offset effect as described in Section 2.3, which occurs when the ambient temperature changes (e.g., by flight altitude). Data in Fig. 4 was detrended to derive the dynamic offset coefficient, which removed all constant offsets that are present when the ambient temperature does not change. For the solar irradiance, we might obtain the "static" offset following Section 2.2. using the original data and assuming that no radiation is present during night. This obviously will amount to the remaining offsets of  $7 \text{ W m}^{-2}$  and  $4 \text{ W m}^{-2}$  that are visible in Fig. 6. However, for two reasons we decided not to follow this approach: A) For the thermal-infrared irradiance this is not possible. B) the "static" offset depends on the difference between dome and sensor temperature, which we don't measure and also can not parameterize. The corrections obtained from the night flight might not be valid for other flights. Contrarily, the dynamic offset depends on the change of temperature only and not on the absolute value of temperatures. This can be quantified during the flights using the sensor temperature.

So it is true, that the remaining deviations shown in Fig. 6 may result from the static thermal offset (Sect. 2.2). In addition, the temperature dependence of the radiometer sensitivity can contribute to the bias. As described in the manuscript, we account for the temperature dependence of the thermopile sensitivities in the data processing. However, the parametrization provided from calibration might not be perfect.

In the revised manuscript, we carefully changed the wording and differentiated between "static" and "dynamic" thermal offset.

The remaining bias to  $F_{sol} = 0 \text{ W m}^{-2}$  is caused by potential static thermal offsets as described in Section 2.2 and other uncertainties such as the radiometric calibration of the pyranometer.

The complex multi-variate fit discussed by the reviewer also refers to the "dynamic" thermal offsets as described in Equation 17 and does not include the "static" offset.

Line 6: it would read better as "an efficient new method".

You are right. We changed the sentence as suggested.

Line 30: should it be "which can be measured directly" Thanks!.

Line 55: should be "Actively stabilized pyranometers"

Thanks! We now tried to consistently use plural throughout the manuscript.

**Line 80: Section 6 is not specifically referenced in the paragraph about the structure of the paper. Did you wish to add that?**

Thanks! This must have deleted by accident during the writing process. We added the reference to Section 6 in the revised manuscript.

**Line 92: should be "In the case of the pyrgeometer"**

We changed to plural "pyrgeometers".

Line 108: I think  $\rho_p$  should be  $\rho_d$  in the  $rho_s \cdot \rho_p << 1$  assumption.

Thanks for finding this tricky typo!

**Line 157: two the's at end of the line**

Thanks!

**Line 244: should be "To enable maintenance"**

Thanks!

**Line 265: What does "one magnitude lower" mean? Does this mean one order of magnitude lower?**

Yes, you are right! We changed the sentence as suggested.

**Line 313: should be "depends"**

Thanks!

**Line 416: The wording at the beginning of this line is unclear.**

We might have used the wrong wording to express, that the general change of the irradiance is caused by the diurnal cycle and that higher frequency oscillations are added. In the revised manuscript, we changed the sentence to:

The uncorrected  $F_{sol}^{\downarrow}$  shows oscillations of different frequency that are superposed to the diurnal cycle.

**Line 508: should be "The data are used by Luebke et al (2022)"**

Thanks! We changed the reference.

**Section 2.3: This paragraph shows the RT model is used in the data, but the justification and uncertainty of this treatment is not well discussed.**

The radiative transfer simulations were not used to replace the measurements, if that is what the reviewer understood. The simulations only provide the relative number of the fraction between direct and solar irradiance, which cannot be measured on the aircraft. This fraction is used to weight the correction of the downward irradiance following the common approach by Bannehr and Schwiesow (1993). The contribution of uncertainties of the direct fraction to the downward radiance strongly depends on solar zenith angle and aircraft attitude. For  $60^{\circ}$  solar zenith angle, roll and pitch angle of  $5^{\circ}$ , 5% uncertainty of the direct fraction amounts to a total uncertainty of less than 1%.

To make this better understandable we changed the section into:

This correction is valid only for the downward direct solar irradiance. Therefore, the relative fractions of direct and diffuse solar radiation in cloud-free conditions are estimated using radiative transfer simulations. ..... For the conditions during ACLOUD, a 5% uncertainty of the simulated fraction of direct radiation amounts to less than 1% uncertainty of the corrected downward irradiance.

**This paragraph also assumes "The upward solar radiation as well as the upward and downward terrestrial radiation were assumed to be isotropic". This is not valid for solar radiation. What's the effect of this assumption?**

This sentence might have been misleading. The point we wanted to make is that upward solar irradiance was not corrected for the aircraft misalignment. This is common procedure because of two reasons. First, a correction would require knowledge on the exact distribution of the radiation field, which is not measured and is difficult to estimate from simulations. Second, the upward radiation is way less anisotropic as the downward radiation (direct solar radiation) and the effects of the aircraft misalignment are little. A perfect isotropic radiation field would cause no effects at all. But it's true that our argumentation was wrong and misleading.

We rephrased this sentence to avoid any misunderstanding.

The upward solar irradiance as well as the upward and downward terrestrial irradiance cannot be corrected for the aircraft attitude. However, these components are characterized by a nearly isotropic radiation field compared to the downward radiation and the effects of a misalignment is minimal for a nearly level sensor (Bucholtz et al. 2008). To limit the remaining uncertainties due to the aircraft movement, measurements with roll and pitch angles exceeding  $\pm 4^{\circ}$  were removed from the data set.

---

## Author Response (AR2)

**Reply to the Editor (AMT-2022-259)**

André Ehrlich[1,★], Martin Zöger[2,★], Andreas Giez[2], Vladyslav Nenakhov[2], Christian Mallaun[2], Rolf Maser[3], Timo Röschenthaler[3], Anna E. Luebke[1], Kevin Wolf[1,a], Bjorn Stevens[4], and Manfred Wendisch[1]

[1]Leipzig Institute for Meteorology, Leipzig University , Germany
[2]German Aerospace Center, Flight Experiments, Oberpfaffenhofen, Germany
[3]enviscope GmbH, Frankfurt am Main, Germany
[4]*Max Planck* Institute for Meteorology, Hamburg, Germany
[a]now at: Institute *Pierre-Simon Laplace*, Sorbonne Université, Paris, France
[★]These authors contributed equally to this work.

**Correspondence:** A. Ehrlich
(a.ehrlich@uni-leipzig.de)

We thank the editor for the additional comments!

The wrong caption of figure 7 is corrected now. Thanks for finding this mistake!

With respect to the spectral ranges of the radiometers and the spectral range of solar and thermal-infrared radiation, we intended to specify different wavelength ranges. To our understanding, the radiometer have a limited spectral sensitivity (CMP22 pyranometer 0.2-3.6 $\mu$m, CGR4 pyrgeometer 4.5-42 $\mu$m). However, the radiometer are calibrated against the entire solar or thermal-infrared spectral range. In the introduction we aim to define the spectral ranges of the measured irradiances (solar and thermal-infrared spectral range). The spectral ranges (0.3-3 $\mu$m and 3-100 $\mu$m) are defined by the WMO Guide to Instruments and Methods of Observation (WMO-No. 8, 2021 edition - Volume I: Measurement of Meteorological Variables) like we introduce in abstract and introduction. Later, when describing the technical parameters of the radiometers, we refer to the spectral sensitivities, which do not cover the entire solar and thermal-infrared spectral range due to sensor and dome characteristics. And in this section we also add, that both radiometer do not cover the entire spectral ranges, which may have consequences for their measurement performance.

We think, it is important to point out, that the measured irradiances refer to the entire spectral ranges. That's why we want to keep this statement in the abstract and introduction. Otherwise, readers may mistakenly do radiative transfer simulations using only the CGR4 or CMP22 spectral range and wonder about discrepancies between simulations and observations.